# A STUDY OF POSTERIOR STABILITY FOR TIME-SERIES LATENT DIFFUSION

## ABSTRACT

Latent diffusion has demonstrated promising results in image generation and permits efficient sampling. However, this framework might suffer from the problem of *posterior collapse* when applied to time series. In this paper, we first show that posterior collapse will reduce latent diffusion to a variational autoencoder (VAE), making it *less expressive*. This highlights the importance of addressing this issue. We then introduce a principled method: *dependency measure*, that quantifies the sensitivity of a recurrent decoder to input variables. Using this tool, we confirm that posterior collapse significantly affects time-series latent diffusion on real datasets, and a phenomenon termed *dependency illusion* is also discovered in the case of shuffled time series. Finally, building on our theoretical and empirical studies, we introduce a new framework that extends latent diffusion and has a stable posterior. Extensive experiments on multiple real time-series datasets show that our new framework is free from posterior collapse and significantly outperforms previous baselines in time series synthesis.

## 1 INTRODUCTION

Latent diffusion (Rombach et al., 2022) has achieved promising performance in image generation and offers significantly higher sampling speeds than standard diffusion models (Ho et al., 2020). However, we find that, when applied to time series data, this framework might suffer from *posterior collapse* (Bowman et al., 2016), an important problem that has garnered significant attention in the literature on autoencoders (Baldi, 2012; Lucas et al., 2019), where the latent variable contains little information about the data and it tends to be ignored by the decoder during conditional generation. In this paper, we aim to provide a systematic analysis on the impact of posterior collapse on latent diffusion and improve this framework based on our analysis.

**Impact analysis of posterior collapse.** We first show that a strictly collapsed posterior reduces the latent diffusion to a variational autoencoder (VAE) (Kingma & Welling, 2013), indicating that this problem makes the framework *less expressive*, even weaker than a vanilla diffusion model. We then introduce a principled method termed *dependency measure*, which quantifies the dependencies of an autoregressive decoder on the latent variable and the input partial time series. Through empirical estimation of these measures, we find that the latent variable has an almost exponentially vanishing impact on the recurrent decoder during the generation process. An example (i.e., the green bar chart) is shown in the upper left subfigure of Fig. 1. More interestingly, the upper right subfigure illustrates a phenomenon we call *dependency illusion*: Even when the time series is randomly shuffled and thus lacks structural dependencies, the decoder of latent diffusion still heavily relies on input observations (instead of the latent variable) for prediction.

**New framework to solve the problem.** We first point out that the root cause of posterior collapse lies in the improper design of latent diffusion, which leads to avoidable KL-divergence regularization and a lack of mechanisms to address the insensitive decoder (Bowman et al., 2016). Building on these findings, we propose a novel framework that extends latent diffusion, allowing the diffusion model and autoencoder to interact more effectively. Specifically, by treating the diffusion process as a form of variational inference, we can eliminate the problematic KL-divergence regularization and permit an unlimited prior distribution for latent variables. To let the decoder be more sensitive to the latent variable, we also apply the diffusion process to simulate a collapsed posterior, imposing

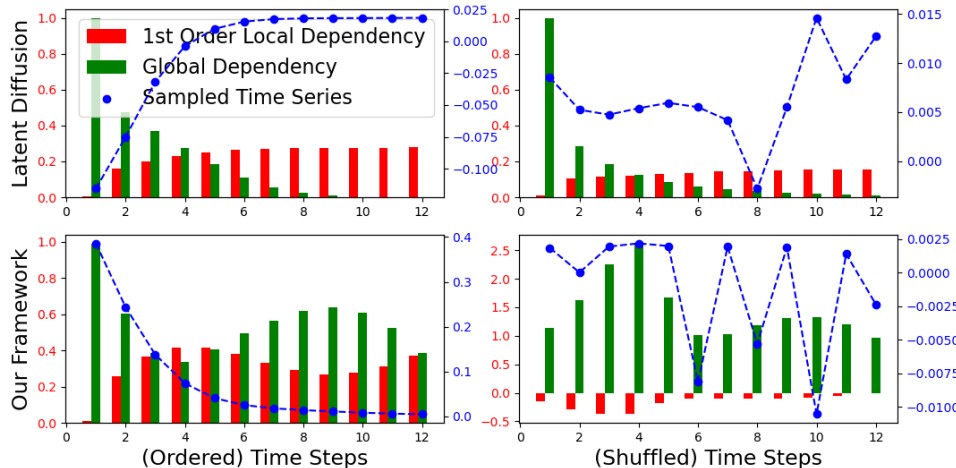

Figure 1: The global and local dependency measures $m_{t,0}, m_{t,t-1}$ (as defined in Sec. 3.2) respectively quantify the impacts of latent variable $\mathbf{z}$ and observation $\mathbf{x}_{t-1}$ on predicting the next one $\mathbf{x}_t$. We can see that the latent variable $\mathbf{z}$ of latent diffusion loses control over the condition generation $p^{\text{gen}}(\mathbf{X} \mid \mathbf{z})$, with *dependency illusion* (as introduced in Sec. 3.3) in the case of shuffled time series. In contrast, our framework has no such symptoms of *posterior collapse*.

a significant penalty on the occurrence of dependency illusion. As demonstrated in the lower two subfigures of Fig. 1, our framework exhibits no signs of posterior collapse, such as the vanishing impact of latent variables over time.

In summary, our paper makes the following contributions:

- We are the first to systematically study *posterior collapse* in latent diffusion, introducing the technique of *dependency measure* (Sec. 3.2) for analysis. We show that the problem renders latent diffusion as inexpressive as a simple VAE (Sec. 3.1) and the latent variable also loses control over time series generation in this case (Sec. 3.3);

- We present a new framework (Sec. 4.2) that improves upon time-series latent diffusion, which eliminates the risky KL-divergence regularization, permits an expressive prior distribution, and features a decoder that is sensitive to the latent variable;

- We have conducted extensive experiments (Sec. 6 and Appendix F) on multiple real time-series datasets, showing that our framework exhibits no symptoms of posterior collapse (or *dependency illusion*) and significantly outperforms previous baselines.

We will publicly release our code upon paper acceptance.

## 2 BACKGROUND: LATENT DIFFUSION

The architecture of latent diffusion consists of two parts: 1) an autoencoder (Baldi, 2012) that maps high-dimensional or structured data into low-dimensional latent variables; 2) a diffusion model (Sohl-Dickstein et al., 2015) that learns the distribution of latent variables.

**Autoencoder.** An implementation for the autoencoder is VAE (Kingma & Welling, 2013). Let $\mathbf{X}$ and $q^{\text{raw}}(\mathbf{X})$ respectively denote the raw data of any form (e.g., pixel matrix) and its distribution. The encoder $\mathbf{f}^{\text{enc}}$ is designed to cast the data $\mathbf{X}$ into a low-dimensional vector $\mathbf{v} = \mathbf{f}^{\text{enc}}(\mathbf{X})$. To get latent variable $\mathbf{z}$, a reparameterization trick is performed as

$$\boldsymbol{\mu} = \mathbf{W}_\mu \mathbf{v}, \quad \boldsymbol{\sigma} = \exp(\mathbf{W}_\sigma \mathbf{v}), \quad \mathbf{z} = \boldsymbol{\mu} + \text{diag}(\boldsymbol{\sigma}) \cdot \boldsymbol{\epsilon}, \boldsymbol{\epsilon} \sim \mathcal{N}(\mathbf{0}, \mathbf{I}), \tag{1}$$

where $\mathbf{W}_\mu, \mathbf{W}_\sigma$ are learnable matrices, and $\text{diag}(\cdot)$ is an operation that casts a vector into a diagonal matrix. The above procedure, which differentially samples a latent variable $\mathbf{z}$ from the posterior $q^{\text{VI}}(\mathbf{z} \mid \mathbf{X}) = \mathcal{N}(\mathbf{z}; \boldsymbol{\mu}, \text{diag}(\boldsymbol{\sigma}^2))$, is called *variational inference* (Blei et al., 2017). The decoder

$\mathbf{f}^{\mathrm{dec}}$ takes latent variable $\mathbf{z}$ as the input to recover the real sample $\mathbf{X}$. In VAE, the decoder output $\mathbf{f}^{\mathrm{dec}}(\mathbf{z})$ is used to parameterize a predefined generation distribution $p^{\mathrm{gen}}(\mathbf{X} \mid \mathbf{z})$.

For training, VAE is optimized in terms of the evidence lower bound (ELBO), an upper bound of the exact negative log-likelihood:

$$\mathcal{L}^{\mathrm{VAE}} = \mathbb{E}_{\mathbf{z} \sim q^{\mathrm{VI}}(\mathbf{z}|\mathbf{X})}[-\ln p^{\mathrm{gen}}(\mathbf{X} \mid \mathbf{z})] + \mathrm{D}_{\mathrm{KL}}(q^{\mathrm{VI}}(\mathbf{z} \mid \mathbf{X}) \parallel p^{\mathrm{prior}}(\mathbf{z})), \qquad (2)$$

where the prior distribution $p^{\mathrm{prior}}(\mathbf{z})$ is commonly set as a standard Gaussian $\mathcal{N}(\mathbf{0}, \mathbf{I})$. The last term of KL divergence leads the prior $p^{\mathrm{prior}}(\mathbf{z})$ to be compatible with the decoder $\mathbf{f}^{\mathrm{dec}}$ for inference, but it is also one cause of *posterior collapse* (Bowman et al., 2016).

**Diffusion model.** An implementation for the diffusion model is DDPM (Ho et al., 2020). The model consists of two Markov chains of $L \in \mathbb{N}^+$ steps. One of them is the diffusion process, which incrementally applies the forward transition kernel:

$$q^{\mathrm{forw}}(\mathbf{z}^i \mid \mathbf{z}^{i-1}) = \mathcal{N}(\mathbf{z}^i; \sqrt{1 - \beta^i}\mathbf{z}^i, \beta^i \mathbf{I}), \qquad (3)$$

where $\beta^i, i \in [1, L]$ is some predefined variance schedule, to the latent variable $\mathbf{z}^0 := \mathbf{z} \sim q^{\mathrm{latent}}(\mathbf{z})$. Here the distribution of latent variable $q^{\mathrm{latent}}(\mathbf{z})$ is defined as $\int q^{\mathrm{VI}}(\mathbf{z} \mid \mathbf{X})q^{\mathrm{raw}}(\mathbf{X})d\mathbf{X}$. The outcomes of this process are a sequence of new latent variables $\{\mathbf{z}^1, \mathbf{z}^2, \cdots, \mathbf{z}^L\}$, with the last one $\mathbf{z}^L$ approximately following a standard Gaussian $\mathcal{N}(\mathbf{0}, \mathbf{I})$ for $L \gg 1$.

The other is the reverse process, which iteratively applies the backward transition kernel,

$$p^{\mathrm{back}}(\mathbf{z}^{i-1} \mid \mathbf{z}^i) = \mathcal{N}(\mathbf{z}^{i-1}; \boldsymbol{\mu}^{\mathrm{back}}(\mathbf{z}^i, i), \sigma^i \mathbf{I}), \quad \boldsymbol{\mu}^{\mathrm{back}}(\mathbf{z}^i, i) = \frac{1}{\sqrt{\alpha^i}}\Big(\mathbf{z}^i - \beta^i \frac{\boldsymbol{\epsilon}^{\mathrm{back}}(\mathbf{z}^i, i)}{\sqrt{1 - \bar{\alpha}^i}}\Big), \quad (4)$$

where $\alpha^i = 1 - \beta^i$, $\bar{\alpha}^i = \prod_{k=1}^i \alpha^k$, $\boldsymbol{\epsilon}^{\mathrm{back}}(\cdot)$ is a neural network, $\mathbf{z}^L$ is an initial sample drawn from $\sim \mathcal{N}(\mathbf{0}, \mathbf{I})$, and $\sigma^i$ is some backward variance schedule. The outcome of this process is a reversed sequence of latent variables $\{\mathbf{z}^{L-1}, \mathbf{z}^{L-2}, \cdots, \mathbf{z}^0\}$, where the last variable $\mathbf{z}^0$ is expected to follow the density distribution of real samples: $q^{\mathrm{latent}}(\mathbf{z}^0)$.

To optimize the diffusion model, common practices adopt a loss function as

$$\mathcal{L}^{\mathrm{DM}} = \mathbb{E}_{i, \mathbf{z}^0, \boldsymbol{\epsilon}}[\|\boldsymbol{\epsilon} - \boldsymbol{\epsilon}^{\mathrm{back}}(\sqrt{\bar{\alpha}^i}\mathbf{z}^0 + \sqrt{1 - \bar{\alpha}^i}\boldsymbol{\epsilon}, i)\|^2], \qquad (5)$$

where $\boldsymbol{\epsilon} \sim \mathcal{N}(\mathbf{0}, \mathbf{I})$, $\mathbf{z}_0 \sim q^{\mathrm{latent}}(\mathbf{z}^0)$, and $i \sim \mathcal{U}\{1, L\}$.

## 3 PROBLEM ANALYSIS

In this section, we first formulate the problem of *posterior collapse* in the framework of time-series latent diffusion and show its significance. Then, we define proper measures that quantify the impact of *posterior collapse* on the models. Finally, we conduct empirical experiments to confirm that time-series diffusion indeed suffers from this problem.

### 3.1 FORMULATION OF POSTERIOR COLLAPSE AND ITS IMPACTS

Let us focus on time series $\mathbf{X} = [\mathbf{x}_1, \mathbf{x}_2, \cdots, \mathbf{x}_T]$, where every observation $\mathbf{x}_t, t \in [1, T]$ is a $D$-dimensional vector and $T$ denotes the number of observations. A potential risk of applying the latent diffusion to time series is the *posterior collapse* (Bowman et al., 2016), which occurs to some autoencoders (Bahdanau et al., 2014), especially VAE (Lucas et al., 2019). Its mathematical formulation in the framework of latent diffusion is as follows.

**Problem formulation.** The posterior of VAE: $q^{\mathrm{VI}}(\mathbf{z} \mid \mathbf{X})$, is collapsed if it reduces to the Gaussian prior $p^{\mathrm{prior}}(\mathbf{z}) = \mathcal{N}(\mathbf{z}; \mathbf{0}, \mathbf{I})$, irrespective of the time-series conditional $\mathbf{X}$:

$$q^{\mathrm{VI}}(\mathbf{z} \mid \mathbf{X}) = p^{\mathrm{prior}}(\mathbf{z}), \forall \mathbf{X} \in \mathbb{R}^{TD}.$$

In this case, the latent variable $\mathbf{z}$ contains no information about time series $\mathbf{X}$, otherwise the posterior distribution $q^{\mathrm{VI}}(\mathbf{z} \mid \mathbf{X})$ would vary depending on different conditionals. Above is a strict definition. In practice, one is mostly faced with a situation where $q^{\mathrm{VI}}(\mathbf{z} \mid \mathbf{X}) \approx p^{\mathrm{prior}}(\mathbf{z})$ and it is still appropriate to say that the posterior collapses.

**Implications of posterior collapse.** A typical symptom of this problem is that, since the latent variable $\mathbf{z}$ carries very limited information of time series $\mathbf{X}$, the trained decoder $\mathbf{f}^{\text{dec}}$ tends to *ignore* this input variable $\mathbf{z}$, which is undesired for conditional generation $p^{\text{gen}}(\mathbf{X} \mid \mathbf{z})$. Besides this empirical finding from previous works, we find that posterior collapse is also significant in its impact on the expressiveness of latent diffusion. Let us first see the below conclusion.

**Proposition 3.1** (Gaussian Latent Variables). *For standard latent diffusion, suppose its posterior $q^{\text{VI}}(\mathbf{z} \mid \mathbf{X})$ is collapsed, then the distribution $q^{\text{latent}}(\mathbf{z})$ of latent variable $\mathbf{z}$ will shape as a standard Gaussian $\mathcal{N}(\mathbf{0}, \mathbf{I})$, which is trivial for the diffusion model to approximate.*

*Proof.* The proof is fully provided in Appendix A. □

In other words, latent variable $\mathbf{z}$ is just Gaussian in the case of posterior collapse. The diffusion model, which is known for approximating complex data distributions (Dhariwal & Nichol, 2021; Li et al., 2024), will in fact become a redundant module. Therefore, posterior collapse reduces latent diffusion to a simple VAE, which also samples latent variable $\mathbf{z}$ from a standard Gaussian $\mathcal{N}(\mathbf{0}, \mathbf{I})$. We conclude that the problem makes latent diffusion *less expressive*.

> **Takeaway**: The problem of *posterior collapse* not only lets the decode $\mathbf{f}^{\text{dec}}$ tend to *ignore* the latent variable $\mathbf{z}$ for conditional generation $p^{\text{gen}}(\mathbf{X} \mid \mathbf{z})$, but also reduces the framework of latent diffusion to VAE, making it *less expressive*.

### 3.2 Introduction of Dependency Measures

It is very intuitive from above that the problem of *posterior collapse* will make the latent variable $\mathbf{z}$ lose control of the decoder $\mathbf{f}^{\text{dec}}$. To make our claim more solid and confirm that the problem happens to time-series diffusion, we introduce some proper measures that quantify the dependencies of decoder $\mathbf{f}^{\text{dec}}$ on various inputs (e.g., variable $\mathbf{z}$).

**Autoregressive decoder.** Consider that decoder $\mathbf{f}^{\text{dec}}$ has an autoregressive structure, which conditions on latent variable $\mathbf{z}$ and prefix $\mathbf{X}_{1:t-1} = [\mathbf{x}_1, \mathbf{x}_2, \cdots, \mathbf{x}_{t-1}]$ to predict the next observation $\mathbf{x}_t$. With abuse of notation, we set $\mathbf{x}_0 = \mathbf{z}$ and formulate the decoder as

$$\mathbf{h}_t = \mathbf{f}^{\text{dec}}(\mathbf{X}_{0:t-1}), \quad \mathbf{X}_{0:t-1} = [\mathbf{x}_0, \mathbf{x}_1, \mathbf{x}_2, \cdots, \mathbf{x}_{t-1}] \tag{6}$$

where the representation $\mathbf{h}_t, t \geq 1$ is linearly projected to multiple parameters (e.g., mean vector and covariance matrix) that determine the distribution $p^{\text{gen}}(\mathbf{x}_t \mid \mathbf{z}, \mathbf{X}_{1:t-1})$ of some family (e.g., Gaussian). Examples of such a decoder include recurrent neural networks (RNN) (Hochreiter & Schmidhuber, 1997) and Transformer (Vaswani et al., 2017). We put the formulation details of these example in Appendix B.

**Dependency measure.** The symptom of posterior collapse is that the decoder $\mathbf{f}^{\text{dec}}$ heavily relies on prefix $\mathbf{X}_{1:t-1}$ (especially the last observation $\mathbf{x}_{t-1}$) to compute the representation $\mathbf{h}_t$, ignoring the guidance of latent variable $\mathbf{x}_0 = \mathbf{z}$. In other words, the variable $\mathbf{z}$ loses control of decoder $\mathbf{f}^{\text{dec}}$ in that situation, which is undesired for conditional generation $p^{\text{gen}}(\mathbf{X} \mid \mathbf{z})$.

Inspired by the technique of integrated gradients (Sundararajan et al., 2017), we present a new tool: *dependency measure*, which quantifies the impacts of latent variable $\mathbf{x}_0 = \mathbf{z}$ and prefix $\mathbf{X}_{1:t-1}$ on decoder $\mathbf{f}^{\text{dec}}$. Specifically, we first set a baseline input $\mathbf{O}_{0:t-1}$ as $[\mathbf{x}_0 = \mathbf{0}, \mathbf{x}_1 = \mathbf{0}, \cdots, \mathbf{x}_{t-1} = \mathbf{0}]$ and denote the term $\mathbf{f}^{\text{dec}}(\mathbf{O}_{0:t-1})$ as $\widetilde{\mathbf{h}}_t$. Then, we parameterize a straight line $\boldsymbol{\gamma}(s) : [0, 1] \to \mathbb{R}^{tD}$ between the actual input $\mathbf{X}_{1:t-1}$ and the input baseline $\mathbf{O}_{0:t-1}$ as

$$\boldsymbol{\gamma}(s) = s\mathbf{X}_{0:t-1} + (1-s)\mathbf{O}_{0:t-1} := [s\mathbf{x}_0, s\mathbf{x}_1, \cdots, s\mathbf{x}_{t-1}]. \tag{7}$$

Applying the chain rule in differential calculus, we have

$$\frac{d\mathbf{f}^{\text{dec}}(\boldsymbol{\gamma}(s))}{ds} = \sum_{j=0}^{t-1} \sum_{k=1}^{k=D} \frac{d\mathbf{f}^{\text{dec}}(\gamma_{j,k}(s))}{d\gamma_{j,k}(s)} \frac{d\gamma_{j,k}(s)}{ds} = \sum_{j=0}^{t-1} \sum_{k=1}^{k=D} x_{j,k} \frac{d\mathbf{f}^{\text{dec}}(\gamma_{j,k}(s))}{d\gamma_{j,k}(s)}, \tag{8}$$

where $\gamma_{j,k}(s)$ denote the $k$-th dimension $s \cdot x_{j,k}$ of the $j$-th vector $s\mathbf{x}_j$ in point $\boldsymbol{\gamma}(s)$. With the above elements, we can define the below measures.

**Definition 3.2** (Dependency Measures). *For an autoregressive decoder $\mathbf{f}^{\text{dec}}$ that conditions on both latent variable $\mathbf{x}_0 = \mathbf{z}$ and the prefix $\mathbf{X}_{1:t-1}$ to compute representation $\mathbf{h}_t$, the dependency measure of every input variable $\mathbf{x}_j, j \in [0, t-1]$ to the decoder is defined as*

$$m_{t,j} = \frac{1}{\|\mathbf{h}_t - \widetilde{\mathbf{h}}_t\|^2} \Big\langle \mathbf{h}_t - \widetilde{\mathbf{h}}_t, \sum_{k=1}^{D} \Big( x_{j,k} \int_0^1 \frac{d\mathbf{f}^{\text{dec}}(\gamma_{j,k}(s))}{d\gamma_{j,k}(s)} ds \Big) \Big\rangle, \tag{9}$$

*where operation $< \cdot, \cdot >$ represents the inner product. In particular, we name $m_{t,0}$ as the global dependency and $m_{t,t-j}, 1 \le j < t$ as the j-th order local dependency.*

We provide the derivation for dependency measure $m_{t,j}$ and detail its relations to integrated gradients in Appendix C. In practice, the integral term can be approximated as

$$\int_0^1 \frac{d\mathbf{f}^{\text{dec}}(\gamma_{j,k}(s))}{d\gamma_{j,k}(s)} ds = \mathbb{E}_{s \in \mathcal{U}\{0,1\}} \Big[ \frac{d\mathbf{f}^{\text{dec}}(\gamma_{j,k}(s))}{d\gamma_{j,k}(s)} \Big] \approx \frac{1}{|\mathcal{S}|} \sum_{s \in \mathcal{S}} \frac{d\mathbf{f}^{\text{dec}}(\gamma_{j,k}(s))}{d\gamma_{j,k}(s)}, \tag{10}$$

where $\mathcal{S}$ is the set of independent samples drawn from uniform distribution $\mathcal{U}\{0,1\}$. According to the law of large numbers (Sedor, 2015), this approximation is unbiased and gets more accurate for a bigger sample set $|\mathcal{S}|$. Notably, the defined measures have the following properties.

**Proposition 3.3** (Signed and Normalization Properties). *The dependency measure $m_{t,j}, \forall j \in [0, t-1]$ is a signed measure and always satisfies $\sum_{j=0}^{t-1} m_{t,j} = 1$.*

*Proof.* The proof is fully provided in Appendix D. $\square$

We can see that the measure $m_{t,j}$ can be either positive or negative, with a normalized sum over the subscript $j$ as 1. If $m_{t,j} \ge 0$, then we say that vector $\mathbf{x}_j$ has a positive impact on the decoder $\mathbf{f}^{\text{dec}}$ for computing representation $\mathbf{h}_j$: the bigger is $m_{t,j}$, the larger is such an impact; Similarly, if $m_{t,j} < 0$, then the vector $\mathbf{x}_j$ has a negative impact on the decoder: the smaller is $m_{t,j}$, the greater is the negative influence. Besides, it is also not hard to understand that there exists a negative impact. For example, the latent variable $\mathbf{z} \sim q^{\text{latent}}(\mathbf{z})$ might be an outlier for the decoder $\mathbf{f}^{\text{dec}}$, which locates at a low-density region in the prior distribution $q^{\text{prior}}(\mathbf{z})$.

**Example Application.** Fig. 1 shows several examples of applying the dependency measures, where each subfigure contains a sample of time series (i.e., blue curve) generated by some model and two types of dependency measures (i.e., red and green bar charts) estimated by Eq. (9). Specifically, every point $\mathbf{x}_t$ in the time series corresponds to a green bar that indicates the global dependency $m_{t,0}$ and a red bar that represents the first-order local dependency $m_{t,t-1}$. In the upper left subfigure, we can see that the positive impact of latent variable $\mathbf{z}$ on the decoder (e.g., $m_{t,0}$) decreases over time and vanishes eventually. From the lower right subfigure, we can even see that some bars (i.e., local dependency $m_{t,t-1}$) are negative, indicating that the variable $\mathbf{x}_{t-1}$ has a negative impact on predicting the next observation $\mathbf{x}_t$.

> **Takeaway**: *Dependency measure $m_{t,j}, 0 \le j < t$ quantifies the impact of latent variable $\mathbf{x}_0 = \mathbf{z}$ or observation $\mathbf{x}_j, j \ge 1$ on the decoder $\mathbf{f}^{\text{dec}}$. This type of impact can be either positive or negative, which is reflected in the value of measure $m_{t,j}$.*

### 3.3 EMPIRICAL DEPENDENCY ESTIMATIONS

We are mainly interested in two types of defined measures: One is the *global dependency $m_{t,0}$*, which estimates the impact of latent variable $\mathbf{x}_0 = \mathbf{z}$ on the decoder $\mathbf{f}^{\text{dec}}$; The other is the *first-order local dependency $m_{t,t-1}$*, which estimates the dependency of decoder $\mathbf{f}^{\text{dec}}$ on the last observation $\mathbf{x}_{t,t-1}$ for computing representation $\mathbf{h}_t$. In this part, we empirically estimate these measures, with the aims to confirm that *posterior collapse* occurs and show its impacts.

**Experiment setup.** Two time-series datasets: WARDS (Alaa et al., 2017b) and MIMIC (Johnson et al., 2016) are adopted. For each dataset, we extract the observations of the first 12 hours, with the top 1 and 5 features that have the highest variances to form univariate and multivariate time

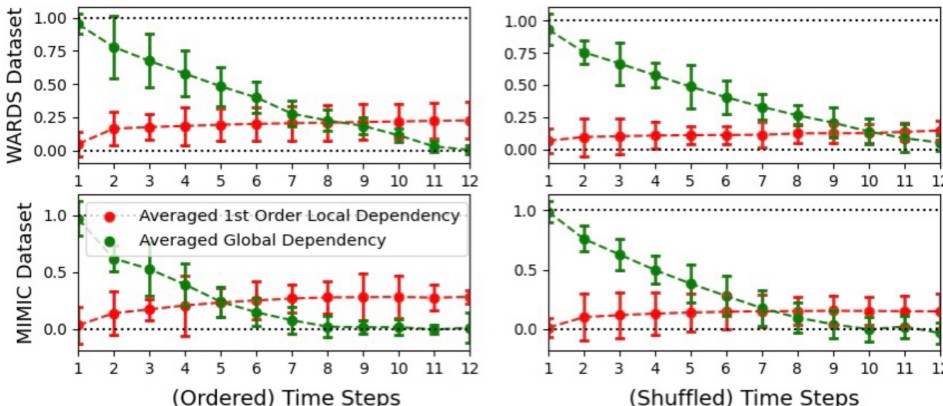

Figure 2: Dependency measures $m_{t,0}, m_{t,t-1}$ averaged over 500 multivariate time series, with 3 standard deviations as the error bars. We can see that the latent variable $\mathbf{z}$ of latent diffusion has a vanishing impact on the decoder $\mathbf{f}^{\mathrm{dec}}$, a typical symptom of *posterior collapse*. We also observe a phenomenon of *dependency illusion* in the case of shuffled time series.

series. To study the case where time series have no structural dependencies, we also try randomly shuffling the time steps of ordered time series. With the prepared datasets, we respectively train latent diffusion models on them and sample time series from the models.

**Insightful results.**  The upper two subfigures of Fig. 1 illustrate the estimated dependencies $m_{t,0}, m_{t,t-1}$ for a single time-series sample $\mathbf{X}$, while Fig. 2 shows the dependency measures averaged over 500 samples. We can see that, for both ordered and shuffled time series, the global dependency $m_{t,0}$ exponentially converges to 0 with increasing time step $t$, indicating that latent variable $\mathbf{z}$ loses control of the generation process of decoder $\mathbf{f}^{\mathrm{dec}}$ and the posterior is *collapsed*. More interestingly, as shown in the right part of Fig. 1, while there is no dependency between adjacent observations $\mathbf{x}_{t-1}, \mathbf{x}_t$ in shuffled time series, we still observe that the first-order measure $m_{t,t-1}$ is significantly different from 0 (e.g., around 0.1 to 0.2). This phenomenon might arise as neural networks overfit and we name it as *dependency illusion*.

> *Takeaway*: Time-series latent diffusion exhibits a typical symptom of *posterior collapse*: latent variable $\mathbf{z}$ has an almost exponentially decreasing impact on generation process $p^{\mathrm{gen}}(\mathbf{X} \mid \mathbf{z})$. More seriously, we observe a phenomenon of *dependency illusion*.

## 4  PROBLEM RETHINKING AND NEW FRAMEWORK

In this section, we first analyze how the framework design of latent diffusion makes it tend to suffer from *posterior collapse*. Then, based on our analysis, we propose a new framework, which extends latent diffusion but addresses the problem.

### 4.1  RISKY DESIGN OF LATENT DIFFUSION

Previous works (Semeniuta et al., 2017; Alemi et al., 2018) have identified two main causes of the problem: *KL-divergence term* and *strong decoder*. For time-series latent diffusion, we will explain as below that those causes indeed exist, but are in fact avoidable.

**Unnecessary regularization.**  The KL-divergence term $\mathrm{D}_{\mathrm{KL}}(q^{\mathrm{VI}}(\mathbf{z} \mid \mathbf{X}) \parallel p^{\mathrm{prior}}(\mathbf{z}))$ in Eq. (2) moves the posterior $q^{\mathrm{VI}}(\mathbf{z} \mid \mathbf{X})$ towards prior $p^{\mathrm{prior}}(\mathbf{z})$, which has the side effect of *posterior collapse* by definition. In essence, this term is tailored for VAE, such that it is valid to sample latent variable $\mathbf{z}$ from the Gaussian prior $p^{\mathrm{prior}}(\mathbf{z})$ for inference. *However, for latent diffusion, the variable $\mathbf{z}$ is sampled from the diffusion model, which can approximate a non-Gaussian prior distribution.* Hence, the interaction between the VAE and diffusion model is not properly designed, which incurs a limited prior $p^{\mathrm{prior}}(\mathbf{z})$ and a risky KL-divergence term $\mathrm{D}_{\mathrm{KL}}(\cdot)$.

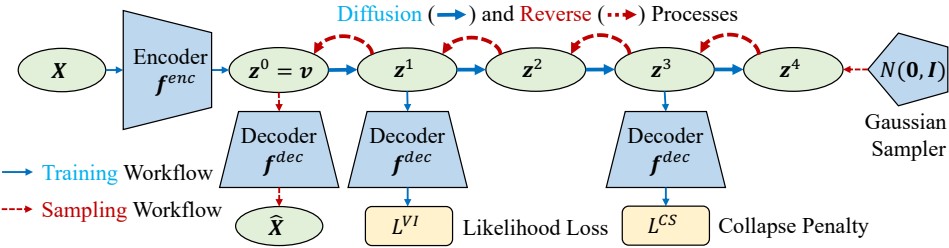

Figure 3: In this example, path $\mathbf{X} \to \mathbf{z}^1$ is the *variational inference* (which gets rid of KL-divergence regularization) and path $\mathbf{X} \to \mathbf{z}^3$ shows the *collapse simulation* (which is to increase the sensitivity of decoder $\mathbf{f}^{\text{dec}}$ to latent variable $\mathbf{z}$). Compared with time-series latent diffusion, our framework is free from *posterior collapse* and has a unlimited prior $p^{\text{prior}}(\mathbf{z})$.

**Unprepared for recurrence.** The strong decoder is also a cause of posterior collapse, which happens to sequence autoencoders (Bowman et al., 2016; Eikema & Aziz, 2019). Time series $\mathbf{X} \in \mathbb{R}^{TD}$ have a clear temporal structure, so the corresponding decoder $\mathbf{f}^{\text{dec}}$ is typically a RNN, which explicitly models the dependency between different observations $\mathbf{x}_i, \mathbf{x}_j, i \neq j$. For predicting the observation $\mathbf{x}_j$, both latent variable $\mathbf{z}$ and previous observation $\mathbf{x}_i, i < j$ are the inputs to the decoder $\mathbf{f}^{\text{dec}}$, so variable $\mathbf{z}$ is possible to be ignored.

Latent diffusion is primarily designed for image generation, with U-net (Ronneberger et al., 2015) as the backbone, which consists of many layers of feedforward neural networks (FNN) (Svozil et al., 1997). For example, convolution neural networks (CNN) (Krizhevsky et al., 2012), self-attention (Vaswani et al., 2017), and MLP (Lu et al., 2017). *These FNN layers are highly sensitive to the input variables, so the original design of latent diffusion lacks a mechanism to address the possible insensitivity*, which is the case of time-series decoder.

> ***Takeaway***: The improper design of latent diffusion is the root cause of *posterior collapse*, which results in the avoidable KL-divergence regularization, limits the form of prior distribution $p^{\text{prior}}(\mathbf{z})$, and lacks a mechanism to handle the insensitive decoder.

## 4.2 NEW FRAMEWORK

In light of previous analyses, we propose a new framework that lets the autoencoder interact with the diffusion model more effectively than latent diffusion. With this better framework design, we can eliminate the KL-divergence term, permit a free from of prior distribution $p^{\text{prior}}(\mathbf{z})$, and increase the sensitivity of decoder $\mathbf{f}^{\text{dec}}$ to latent variable $\mathbf{z}$.

Importantly, we notice a conclusion (Ho et al., 2020) for the diffusion process (i.e., Eq. (3)):

$$q^{\text{forw}}(\mathbf{z}^i \mid \mathbf{z}^0) = \mathcal{N}(\mathbf{z}^i; \sqrt{\bar{\alpha}^i}\mathbf{z}^0, (1 - \bar{\alpha}^i)\mathbf{I}), \tag{11}$$

where the coefficient $\bar{\alpha}^i$ monotonically decreases from 1 to approximately 0 for $i \in [0, L]$. In this sense, suppose the initial variable $\mathbf{z}^0$ is set as $\mathbf{v} = \mathbf{f}^{\text{enc}}(\mathbf{X})$, then we can infer that the random variable $\mathbf{z}^i \sim q^{\text{forw}}(\mathbf{z}^i \mid \mathbf{z}^0)$ contains $\bar{\alpha}^i \times 100\%$ information about the vector $\mathbf{v}$, with $(1 - \bar{\alpha}^i) \times 100\%$ pure noise. For $i \to 0$, the diffusion process is similar to the *variational inference* (i.e., Eq. (1)) of VAE, adding slight Gaussian noise to the encoder output $\mathbf{v}$. For $i \to T$, the variable $\mathbf{z}^i$ simulates the problem of *posterior collapse* since $q^{\text{forw}}(\mathbf{z}^i \mid \mathbf{z}^0) \approx \mathcal{N}(\mathbf{z}^i; \mathbf{0}, \mathbf{I})$.

**Diffusion process as variational inference.** Considering the above facts, we first treat the starting few iterations of the diffusion process as the *variational inference*. Specifically, with a fixed small integer $N \ll L$, we sample a number $i$ from uniform distribution $\mathcal{U}\{0, N\}$ and let the diffusion process convert the encoder output $\mathbf{v} = \mathbf{f}^{\text{enc}}(\mathbf{X})$ into the latent variable:

$$\mathbf{z} = \mathbf{z}^i \sim q^{\text{forw}}(\mathbf{z}^i \mid \mathbf{z}^0), \mathbf{z}^0 = \mathbf{v}. \tag{12}$$

In terms of the formerly defined generation distribution $p^{\text{gen}}(\mathbf{X} \mid \mathbf{z})$ (parameterized by the decoder $\mathbf{f}^{\text{dec}}$), a negative log-likelihood loss $\mathcal{L}^{\text{VI}}$ is incurred as

$$\mathcal{L}^{\text{VI}} = \mathbb{E}_{i \sim \mathcal{U}\{0, N\}, \mathbf{z}^0}[-\bar{\alpha}^{\gamma i} \ln p^{\text{gen}}(\mathbf{X} \mid \mathbf{z} = \mathbf{z}^i)], \tag{13}$$

| **Algorithm 1** Training | **Algorithm 2** Sampling |
|---|---|
| 1: **repeat** | 1: $\mathbf{z}_L \sim p^{\text{back}}(\mathbf{z}_L) = \mathcal{N}(\mathbf{0}, \mathbf{I})$ |
| 2:     Sample time series $\mathbf{X}$ from the dataset | 2: Set stop time: $i \sim \mathcal{U}\{0, N\}$ |
| 3:     Representation encoding: $\mathbf{v} = \mathbf{f}^{\text{enc}}(\mathbf{X})$ | 3: **for** $l = L, L-1, \ldots, i+1$ **do** |
| 4:     $\mathbf{z}^j \sim q^{\text{forw}}(\mathbf{z}^j \mid \mathbf{z}^0 = \mathbf{v}), j \sim \mathcal{U}\{0, N\}$ | 4:     $\mathbf{z}^{l-1} \sim p^{\text{back}}(\mathbf{z}^{l-1} \mid \mathbf{z}^l)$ |
| 5:     $\widehat{\mathcal{L}}^{\text{VI}} = -\bar{\alpha}^{\gamma j} \ln p^{\text{gen}}(\mathbf{X} \mid \mathbf{z} = \mathbf{z}^j)$ | 5: **end for** |
| 6:     $i \sim \mathcal{U}\{j, L\}, \boldsymbol{\epsilon} \sim \mathcal{N}(\mathbf{0}, \mathbf{I})$ | 6: Conditional generation: $p^{\text{gen}}(\widehat{\mathbf{X}} \mid \mathbf{z} = \mathbf{z}^i)$ |
| 7:     $\widehat{\mathcal{L}}^{\text{DM}} = \|\boldsymbol{\epsilon} - \boldsymbol{\epsilon}^{\text{back}}(\sqrt{\bar{\alpha}^i}\mathbf{z}^j + \sqrt{\cdot}\boldsymbol{\epsilon}, i)\|^2$ | 7: **return** Time series $\widehat{\mathbf{X}}$ |
| 8:     $\mathbf{z}^k \sim q^{\text{forw}}(\mathbf{z}^k \mid \mathbf{z}^0 = \mathbf{v}), k \sim \mathcal{U}\{M, L\}$ | |
| 9:     $\widehat{\mathcal{L}}^{\text{CS}} = (1 - \bar{\alpha}^{\lceil \frac{k}{\eta} \rceil}) \ln p^{\text{gen}}(\mathbf{X} \mid \mathbf{z} = \mathbf{z}^k)$ | |
| 10:    Gradient descent with $\nabla(\widehat{\mathcal{L}}^{\text{VI}} + \widehat{\mathcal{L}}^{\text{DM}} + \widehat{\mathcal{L}}^{\text{CS}})$ | |
| 11: **until** converged | |

where $\gamma \in \mathbb{N}^+, \gamma N \leq L$ is a hyper-parameter, with the aim to reduce the impact of a very noisy latent variable $\mathbf{z}$. As multiplier $\gamma$ increases, the weight $\bar{\alpha}^{\gamma i}$ decreases.

Similar to VAE, the variational inference in our framework also leads the latent variable $\mathbf{z}$ to be *smooth* (Bowman et al., 2016) in its effect on decoder $\mathbf{f}^{\text{dec}}$. However, our framework is free from the KL-divergence term $\text{D}_{\text{KL}}(q^{\text{VI}}(\mathbf{z} \mid \mathbf{X}) \| p^{\text{prior}}(\mathbf{z}))$ of VAE (i.e., one cause of the *posterior collapse*), since we can facilitate $\mathbf{z} \sim q^{\text{latent}}(\mathbf{z})$ at test time through applying the reverse process of the diffusion model (i.e., Eq. (4)) to sample variable $\mathbf{z}^i, i \in [0, N]$.

**Diffusion process for collapse simulation.** Then, we apply the last few iterations of the diffusion process to simulate *posterior collapse*, with the purposes of increasing the impact of latent variable $\mathbf{z}$ on conditional generation $p^{\text{gen}}(\mathbf{X} \mid \mathbf{z})$ and reducing *dependency illusion*.

Following our previous *variational inference*, we set $\mathbf{z}^0 = \mathbf{f}^{\text{enc}}(\mathbf{X})$ and apply the diffusion process to cast the initial variable $\mathbf{z}^0$ into a highly noisy variable $\mathbf{z}^i, i \to L$. Considering that the variable $\mathbf{z}^i$ contains little information about the encoder output $\mathbf{f}^{\text{enc}}(\mathbf{X})$, it is unlikely that the decoder $\mathbf{f}^{\text{dec}}$ can recover time series $\mathbf{X}$ from variable $\mathbf{z}^i$, otherwise there is *posterior collapse* or *dependency illusion*. In this sense, we have the following regularization:

$$\mathcal{L}^{\text{CS}} = \mathbb{E}_{i \sim \mathcal{U}\{M, L\}, \mathbf{z}^i}[(1 - \bar{\alpha}^{\lceil \frac{i}{\eta} \rceil}) \ln p^{\text{gen}}(\mathbf{X} \mid \mathbf{z} = \mathbf{z}^i)], \tag{14}$$

which penalizes the model for having a high conditional density $p^{\text{gen}}(\mathbf{X} \mid \mathbf{z})$ for non-informative latent variable $\mathbf{z} = \mathbf{z}^i, i \in [M, L]$. Here $M \in \mathbb{N}^+$ is close to $L$, $\lceil \cdot \rceil$ is the ceiling function, and $\eta \geq 1$ is set to reduce the impact of informative variable $\mathbf{z}^i$.

For a *strong decoder* $\mathbf{f}^{\text{dec}}$, such as long short-term memory (LSTM), the regularization $\mathcal{L}^{\text{CS}}$ will impose a heavy penalty if the decoder solely relies on previous observations $\{\mathbf{x}_k \mid k < j\}$ to predict an observation $\mathbf{x}_j$. In that situation, a high prediction probability will be assigned to the observation $\mathbf{x}_j$ even if the latent variable $\mathbf{z}$ contains very limited information about the raw data $\mathbf{X}$.

**Training, inference, and running times.** Our framework is very different from the latent diffusion in both training and inference. We depict the workflows of our framework in Fig. 3, with the training and sampling procedures respectively placed in Algorithm 1 and Algorithm 2. For training, the key points are to compute three loss functions: $\widehat{\mathcal{L}}^{\text{VI}}$ for likelihood maximization, $\widehat{\mathcal{L}}^{\text{DM}}$ for training diffusion models, and $\widehat{\mathcal{L}}^{\text{CS}}$ for collapse regularization. For inference, the main difference is that the stopping time of the backward process is not 0, but a random variable. From these pseudo codes, we can see that our framework is almost as efficient as latent diffusion. We provide an in-depth analysis and empirical experiments about the running times of our framework in Appendix F.2.

## 5 RELATED WORK

Besides latent diffusion, our paper is also related to previous works that aim to mitigate the problem of *posterior collapse* for VAE (Kingma & Welling, 2013). We collect three main types of such

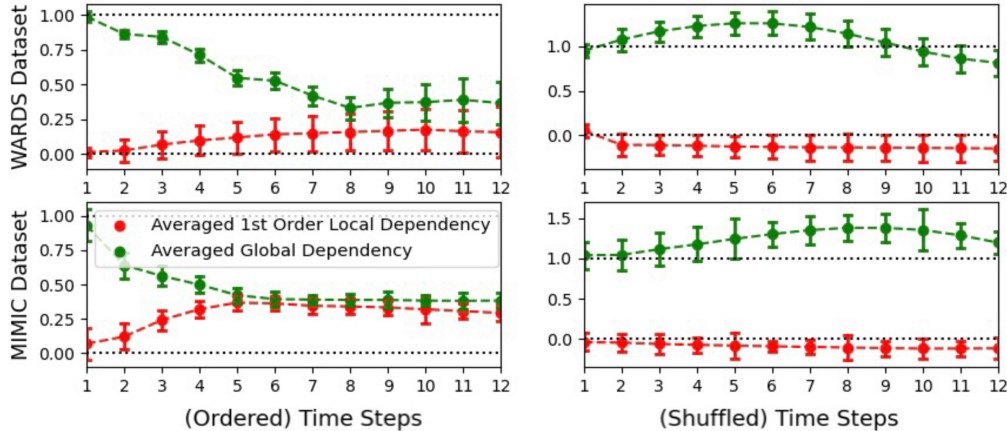

Figure 4: The results of averaged dependency measures and error bars for our framework, which should be compared with those (e.g., Fig. 2) of latent diffusion, showing that our framework has a *stable posterior* and is without *dependency illusion*.

methods and apply them to improve the VAE of latent diffusion, with the corresponding experiment results shown in Table 1 of Sec. 6 and Table 4 of Appendix F.3. In the following, we briefly introduce those baselines and explain their limitations.

**KL annealing.** This method (Bowman et al., 2016) assigns an adaptive weight to control the effect of KL-divergence term, so that VAE is unlikely to fall into the local optimum of posterior collapse at the initial optimization stage. While this method indeed mitigates the problem, it still cannot fully eliminate the negative impact of the risky KL-divergence regularization.

**Decoder weakening.** A representative method in this class is Variable Masking (Semeniuta et al., 2017), which randomly masks input observations to the autoregressive decoder, such that the decoder is forced to rely more on the latent variable for predicting the next observation. However, this method will make the model less expressive since the decoder is weakened.

**Skip connections.** With the aim to improve the impact of latent variables on the recurrent decoder, this approach (Dieng et al., 2019) directly feeds the latent variable into the decoder at every step, not only at the first step. However, the latent variable in that case acts as a constant input signal at every time step, so the decoder will still tend to ignore this redundant information.

Compared with the above baselines, our framework can address the problem of *posterior collapse* and is free from their side effects (e.g., less expressive decoder). The experiment results in Sec. 6 and Appendix F confirm that our framework indeed performs better in practice.

## 6 EXPERIMENTS

We have conducted extensive experiments to verify that our framework is free from *posterior collapse* and outperforms latent diffusion (with or without previous baselines) in terms of time series generation. Due to the limited space, some other important empirical studies are put in the appendix, including *more time-series datasets and another evaluation metric in Appendix F.3*, diverse data modalities (e.g., text) in Appendix F.4, ablation studies in Appendix F.1, and the study of running times in Appendix F.2. The experiment setup is also placed in Appendix E.

### 6.1 STABLE POSTERIOR OF OUR FRAMEWORK

To show that our framework has a non-collapsed posterior $q^{\text{VI}}(\mathbf{z} \mid \mathbf{X})$, we follow the same experiment setup (e.g., datasets) as Sec. 3.3 and average the dependency measures $m_{t,0}, m_{t,t-1}$ over $500$ sampled time series. The results are illustrated in Fig. 4. For ordered time series in the left two

| Model | Backbone | MIMIC | WARDS | Earthquakes |
|---|---|---|---|---|
| Latent Diffusion | LSTM | 5.19 | 7.52 | 5.87 |
| Latent Diffusion w/ KL Annealing | LSTM | 4.28 | 5.74 | 3.88 |
| Latent Diffusion w/ Variable Masking | LSTM | 4.73 | 6.01 | 4.26 |
| Latent Diffusion w/ Skip Connections | LSTM | 3.91 | 4.95 | 3.74 |
| Our Framework | LSTM | **2.29** | **3.16** | **2.67** |
| Latent Diffusion | Transformer | 5.02 | 7.46 | 5.91 |
| Latent Diffusion w/ KL Annealing | Transformer | 4.31 | 5.54 | 3.51 |
| Latent Diffusion w/ Variable Masking | Transformer | 4.42 | 5.97 | 4.45 |
| Latent Diffusion w/ Skip Connections | Transformer | 3.75 | 4.67 | 3.69 |
| Our Framework | Transformer | **2.13** | **3.01** | **2.49** |

Table 1: Wasserstein distances of different models on three widely used time-series datasets. The lower the distance metric, the better the generation quality. *More results from other time-series datasets, with another evaluation metric, are placed in Table 4 of Appendix F.3.*

subfigures, we can see that, while the global dependency $m_{t,0}$ still decreases with increasing time step $t$, it converges into a value around $0.5$, which is also a bit higher than the converged first-order local dependency $m_{t,t-1}$. These results indicate that latent variable $\mathbf{z}$ in our framework maintains its control of decoder $\mathbf{f}^{\mathrm{dec}}$ during the whole conditional generation process $p^{\mathrm{gen}}(\mathbf{X} \mid \mathbf{z})$.

For shuffled time series in the right two subfigures, we can see that the global dependency $m_{t,0}$ is always around or above $1$, and the local dependency $m_{t,t-1}$ is negative most of the time. These results indicate that the decoder $\mathbf{f}^{\mathrm{dec}}$ only relies on latent variable $\mathbf{z}$ and the context $\mathbf{x}_{t-1}$ even has a negative impact on conditional generation $p^{\mathrm{gen}}(\mathbf{X} \mid \mathbf{z})$, suggesting our framework is without *dependency illusion*. Based on all our findings, we conclude that: compared with latent diffusion (Fig. 2), our framework is free from the effects of posterior collapse (e.g., *strong decoder*).

### 6.2 Performances in Time Series Generation

In this part, we aim to verify that our framework outperforms latent diffusion in terms of time series generation, which is intuitive since our framework is free from *posterior collapse*. We also include some other methods that are proposed by previous works to mitigate the problem, including KL annealing (Fu et al., 2019a), variable masking (Bowman et al., 2016), and skip connections (Dieng et al., 2019). We adopt the Wasserstein distances (Bischoff et al., 2024) as the metric.

The experiment results on three commonly used time-series datasets are shown in Table 1. From the results, we can see that, regardless of the used dataset and the backbone of autoencoder, our framework significantly outperforms latent diffusion and the baselines, which strongly confirms our intuition. For example, with the backbone of Transformer, our framework achieves 2.53 points lower than latent diffusion w/ KL Annealing on the WARDS dataset.

### 7 Conclusion

In this paper, we provide a solid analysis of the negative impacts of *posterior collapse* on time-series latent diffusion and introduce a new framework that is free from this problem. For our analysis, we begin with a theoretical insight, showing that the problem will reduce latent diffusion to VAE, rendering it *less expressive*. Then, we introduce a useful tool: *dependency measure*, which quantifies the impacts of various inputs on an autoregressive decoder. Through empirical dependency estimation, we show that the latent variable has a vanishing impact on the decoder and find that latent diffusion exhibits a phenomenon of *dependency illusion*. Compared with standard latent diffusion, our framework gets rid of the risky KL-divergence regularization, permits an unlimited prior distribution, and lets the decoder be sensitive to the latent variable. Extensive experiments on multiple real-world time-series datasets show that our framework has no symptoms of posterior collapse and notably outperforms the baselines in terms of time series generation.

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

# Appendix

# A   THE IMPACT OF POSTERIOR COLLAPSE

Under the assumption of *posterior collapse*, the below equality:

$$q^{\mathrm{VI}}(\mathbf{z} \mid \mathbf{X}) = p^{\mathrm{prior}}(\mathbf{z}) = \mathcal{N}(\mathbf{z}; \mathbf{0}, \mathbf{I}), \tag{15}$$

holds for any latent variable $\mathbf{z} \in \mathbb{R}^D$ and any conditional $\mathbf{X} \in \mathbb{R}^{TD}$. Then, note that

$$
\begin{aligned}
q^{\mathrm{latent}}(\mathbf{z}) &= \int q^{\mathrm{VI}}(\mathbf{z} \mid \mathbf{X}) q^{\mathrm{raw}}(\mathbf{X}) d\mathbf{X} = \int \mathcal{N}(\mathbf{z}; \mathbf{0}, \mathbf{I}) q^{\mathrm{raw}}(\mathbf{X}) d\mathbf{X} \\
&= \mathcal{N}(\mathbf{z}; \mathbf{0}, \mathbf{I}) \int q^{\mathrm{raw}}(\mathbf{X}) d\mathbf{X} = \mathcal{N}(\mathbf{z}; \mathbf{0}, \mathbf{I}),
\end{aligned}
\tag{16}
$$

which is exactly our claim.

# B   RECURRENT ENCODERS

We mainly implement the backbone of decoder $\mathbf{f}^{\mathrm{dec}}$ as LSTM (Hochreiter & Schmidhuber, 1997) or Transformer (Vaswani et al., 2017). In the former case, we apply the latent variable $\mathbf{z}$ to initialize LSTM and condition it on prefix $\mathbf{X}_{1:t-1}$ to compute the representation $\mathbf{h}_t$. Formally, the LSTM-based decoder $\mathbf{f}^{\mathrm{dec}}$ is as

$$
\begin{cases}
\mathbf{s}_t = \mathrm{LSTM}(\mathbf{s}_{t-1}, \mathbf{x}_{t-1}), \forall t \geq 1 \\
\mathbf{h}_t = \mathbf{W}_f^2 \tanh(\mathbf{W}_f^1 \mathbf{s}_t)
\end{cases}, \tag{17}
$$

where $\mathbf{s}_t$ is the state vector of LSTM and $\mathbf{W}_f^2, \mathbf{W}_f^1$ are learnable matrices. In particular, for the corner case $t = 1$, we fix $\mathbf{s}_0, \mathbf{x}_0$ as zero vectors.

In the later case, we just treat latent variable $\mathbf{z}$ as $\mathbf{x}_0$. Therefore, we have

$$
\begin{cases}
[\mathbf{s}_{t-1}, \mathbf{s}_{s-2}, \cdots, \mathbf{s}_0] = \mathrm{Transformer}(\mathbf{x}_{t-1}, \mathbf{x}_{t-2}, \cdots, \mathbf{x}_0) \\
\mathbf{h}_t = \mathbf{W}_f^2 \tanh(\mathbf{W}_f^1 \mathbf{s}_{t-1})
\end{cases}, \tag{18}
$$

where the subscript alignment results from self-attention mechanism.

# C   DERIVATION OF DEPENDENCY MEASURES

Integrated gradient (Sundararajan et al., 2017) is a very effective method of feature attributions. Our proposed dependency measures can be regarded as its extension to the case of sequence data and vector-valued neural networks. In the following, we provide the derivation of dependency measures.

For the computation $\mathbf{h}_t = \mathbf{f}^{\mathrm{dec}}(\mathbf{X}_{0:t-1})$, suppose the output of decoder $\mathbf{f}^{\mathrm{dec}}$ at origin $\mathbf{O}_{0:t-1}$ is $\widetilde{\mathbf{h}}_t$, then we apply the fundamental theorem of calculus as

$$\mathbf{h}_t - \widetilde{\mathbf{h}}_t = \int_0^1 \frac{d\mathbf{f}^{\mathrm{dec}}(\boldsymbol{\gamma}(s))}{ds} ds, \tag{19}$$

where $\boldsymbol{\gamma}(s)$ is a straight line connecting the origin $\mathbf{O}_{0:t-1}$ and the input $\mathbf{X}_{0:t-1}$ as $\boldsymbol{\gamma}(s) = s\mathbf{X}_{0:t-1} + (1-s)\mathbf{O}_{0:t-1}$. Based on the chain rule, the above equality can be expanded as

$$
\begin{aligned}
\mathbf{h}_t - \widetilde{\mathbf{h}}_t &= \int_0^1 \sum_{j=0}^{t-1} \sum_{k=1}^{k=D} \frac{d\mathbf{f}^{\mathrm{dec}}(\gamma_{j,k}(s))}{d\gamma_{j,k}(s)} \frac{d\gamma_{j,k}(s)}{ds} ds \\
&= \sum_{j=0}^{t-1} \left( \int_0^1 \sum_{k=1}^{k=D} x_{j,k} \frac{d\mathbf{f}^{\mathrm{dec}}(\gamma_{j,k}(s))}{d\gamma_{j,k}(s)} ds \right),
\end{aligned}
\tag{20}
$$

where $\gamma_{j,k}(s)$ denote the $k$-th dimension $s \cdot x_{j,k}$ of the $j$-th vector $s\mathbf{x}_j$ in point $\boldsymbol{\gamma}(s)$. Intuitively, every term inside the outer sum operation $\sum_{j=0}^{t-1}$ represents the additive contribution of variable $\mathbf{x}_j$ (to the output difference $\mathbf{h}_t - \widetilde{\mathbf{h}}_t$) along the integral line $\boldsymbol{\gamma}(s)$.

| Model | Backbone | $N$ for $\mathcal{L}^{\mathrm{VI}}$ | $M$ for $\mathcal{L}^{\mathrm{CS}}$ | Diffusion Iterations $L$ | MIMIC | WARDS |
|---|---|---|---|---|---|---|
| Latent Diffusion | Transformer | − | − | 1000 | 5.02 | 7.46 |
| LD w/ Skip Connections | Transformer | − | − | 1000 | 3.75 | 4.67 |
| Our Framework | Transformer | 50 | 100 | 1000 | **2.13** | **3.01** |
| Our Framework | Transformer | 50 | 50 | 1000 | 2.59 | 3.32 |
| Our Framework | Transformer | 50 | 150 | 1000 | 2.71 | 3.46 |
| Our Framework | Transformer | 50 | 200 | 1000 | 2.83 | 3.75 |
| Our Framework | Transformer | 10 | 100 | 1000 | 2.31 | 3.16 |
| Our Framework | Transformer | 100 | 100 | 1000 | 2.38 | 3.24 |
| Our Framework | Transformer | 150 | 100 | 1000 | 2.75 | 3.41 |

Table 2: Ablation studies of the hyper-parameters $N, M$, which are respectively used in the estimations of likelihood loss $\mathcal{L}^{\mathrm{VI}}$ and collapse penalty $\mathcal{L}^{\mathrm{CS}}$. Here LD is short for latent diffusion and the symbol − means "Not Applicable".

To simplify the notation, we denote the mentioned term as

$$\mathbf{m}_{t,j} = \int_0^1 \sum_{k=1}^{k=D} x_{j,k} \frac{d\mathbf{f}^{\mathrm{dec}}(\gamma_{j,k}(s))}{d\gamma_{j,k}(s)} ds. \tag{21}$$

Since $\mathbf{m}_{t,j}$ is a vector, we map the new term to a scalar and re-scale it as

$$m_{t,j} = \frac{< \mathbf{m}_{t,j}, \mathbf{h}_t - \widetilde{\mathbf{h}}_t >}{< \mathbf{h}_t - \widetilde{\mathbf{h}}_t, \mathbf{h}_t - \widetilde{\mathbf{h}}_t >}, \tag{22}$$

which is exactly our definition of the dependency measure.

## D PROPERTIES OF OF DEPENDENCY MEASURES

Firstly, in terms of Eq. (22), it is obvious that the dependency measure $m_{t,j}$ is signed: the measure can be either positive or negative. Then, based on Eq. (20), we have

$$\mathbf{h}_t - \widetilde{\mathbf{h}}_t = \sum_{j=0}^{t-1} \mathbf{m}_{t,j}. \tag{23}$$

By taking an inner product with the vector $\mathbf{h}_t - \widetilde{\mathbf{h}}_t$ at both sides, we get

$$< \mathbf{h}_t - \widetilde{\mathbf{h}}_t, \mathbf{h}_t - \widetilde{\mathbf{h}}_t >=< \sum_{j=0}^{t-1} \mathbf{m}_{t,j}, \mathbf{h}_t - \widetilde{\mathbf{h}}_t >= \sum_{j=0}^{t-1} < \mathbf{m}_{t,j}, \mathbf{h}_t - \widetilde{\mathbf{h}}_t > . \tag{24}$$

By rearranging the term, we finally arrive at

$$1 = \sum_{j=0}^{t-1} \frac{< \mathbf{m}_{t,j}, \mathbf{h}_t - \widetilde{\mathbf{h}}_t >}{< \mathbf{h}_t - \widetilde{\mathbf{h}}_t, \mathbf{h}_t - \widetilde{\mathbf{h}}_t >} = \sum_{j=0}^{t-1} m_{t,j}, \tag{25}$$

which is exactly our claim.

## E EXPERIMENT DETAILS

We have adopted three widely used time-series datasets for both analysis and model evaluation, including MIMIC (Johnson et al., 2016), WARDS (Alaa et al., 2017a), and Earthquakes (U.S. Geological Survey, 2020). The setup of the first two datasets are introduced in Sec. 3.3. For MIMIC, we specially simplify it into a version of univariate time series for the illustration purpose, which is only used in the experiments shown in Fig. 1. All other experiments are about multivariate time series. For the Earthquakes dataset, it is about the location and time of all earthquakes in Japan from 1990 to 2020 with magnitude of at least 2.5 from U.S. Geological Survey (2020). We follow the same preprocessing procedure for this dataset as Li (2023).

| Method | Training Time | Inference Time |
|---|---|---|
| Latent Diffusion | 2hr 10min | 5min 12s |
| Our Framework | 2hr 50min | 5min 17s |

Table 3: Comparison of Training and Inference Times on the MIMIC dataset.

| Method | Backbone | Retail | Energy |
|---|---|---|---|
| Latent Diffusion | Transformer | 0.037 | 0.052 |
| Latent Diffusion w/ Skip Connections | Transformer | 0.033 | 0.043 |
| Our Framework | Transformer | **0.025** | **0.031** |
| Latent Diffusion | LSTM | 0.041 | 0.057 |
| Latent Diffusion w/ Skip Connections | LSTM | 0.035 | 0.047 |
| Our Framework | LSTM | **0.027** | **0.033** |

Table 4: Comparison on two new time-series datasets, with another metric: MMD.

We use almost the same model configurations for all experiments. The diffusion models are parameterized by a standard U-net (Ronneberger et al., 2015), with $L = 1000$ diffusion iterations and hidden dimensions $\{128, 64, 32\}$. The hidden dimensions of autoencoders and latent variables are fixed as $128$. The conditional distribution $p^{\text{gen}}(\mathbf{X} \mid \mathbf{z})$ is parameterized as a Gaussian, with learnable mean vector and diagonal covariance matrix functions. For our framework, $N, M$ are respectively selected as $50, 100$, with $\gamma = 2$ and $\eta = 1$. We also apply dropout with a ratio of $0.1$ to most layers of neural networks. We adopt Adam algorithm (Kingma & Ba, 2015) with the default hyper-parameter setting to optimize our model. For Table 1 and Table 2, every number is averaged over 10 different random seeds, with a standard deviation less than $0.05$. For the computing resources, all our models can be trained on 1 NVIDIA Tesla V100 GPU within 10 hours.

# F   ADDITIONAL EXPERIMENTS

Due to the limited space of our main text, we put the results of some minor experiments here in the appendix. Notably, we will adopt more datasets and another evaluation metric.

## F.1   ABLATION STUDIES

We have conducted ablation studies to verify that our hyper-parameter selections $N = 50, M = 100$ are optimal. The experiment results are shown in Table 2. For both $N$ and $M$, either increasing or decreasing their values results in worse performance on the two datasets.

## F.2   STUDY ON RUNNING TIMES

Our framework only incurs a minor increase in training time and enjoys the same inference speed as the latent diffusion. For training, while our framework will run the decoder a second time for collapse simulation $\mathcal{L}^{\text{CS}}$, it can be made in parallel with the first run of decoder $f^{\text{dec}}$ for likelihood computation $\mathcal{L}^{\text{VI}}$. Therefore, the training is still efficient on GPU devices. Our framework also has a different way of variational inference to infer latent variable $\mathbf{z}$ from data $\mathbf{X}$. However, it admits a closed-form solution and is thus as efficient as the reparameterization trick of latent diffusion. For inference, our framework has no difference from the latent diffusion: sampling the latent variable $\mathbf{z}$ with the reverse diffusion process and running the decoder $f^{\text{dec}}$ in one shot.

To show the running times in practice, we perform an experiment on the MIMIC dataset as shown in Table 3. We can see that our framework indeed only has a minor increase for training. Given that our framework is free from posterior collapse and delivers better generation performances, this slight time investment is well worth it.

## F.3   MORE DATASETS AND ANOTHER EVALUATION METRIC

We conduct additional experiments on 2 more public UCI time-series datasets (Bay et al., 2000): Retail and Energy, with another widely used evaluation metric: maximum mean discrepancy

| Method | Backbone | ATIS | SNIPS |
|---|---|---|---|
| Latent Diffusion | Transformer | 37.12 | 59.36 |
| Latent Diffusion w/ Skip Connections | Transformer | 40.56 | 65.41 |
| Our Framework | Transformer | **51.73** | **78.12** |
| Latent Diffusion | LSTM | 35.38 | 55.72 |
| Latent Diffusion w/ Skip Connections | LSTM | 39.27 | 60.31 |
| Our Framework | LSTM | **48.46** | **71.45** |

Table 5: Performance comparison on two text datasets, with BLEU as the metric.

| Method | Backbone | CIFAR-10 |
|---|---|---|
| Latent Diffusion | U-Net | 3.91 |
| Latent Diffusion w/ KL Annealing | U-Net | 3.87 |
| Our Framework | U-Net | **3.85** |

Table 6: Performance comparison on an image dataset, with FID as the metric.

(MMD) (Dziugaite et al., 2015). Lower MMD scores indicate better generative models. From the results shown in Table 4. We can see that our framework still significantly outperforms the baselines in terms of all the new benchmarks. For example, with LSTM as the backbone, our framework achieves a score that is $29.79\%$ lower than Skip Connections on the Energy dataset.

### F.4 MORE DATA MODALITIES

While our paper primarily focused on time series data, our framework is generally applicable to other types of data, including your mentioned text and images.

**Experiment on text data.** For text data, considering that natural language sentences exhibits a sequential structure similar to time series, it is intuitive that the posterior of text latent diffusion might also collapse. This intuition is supported by many evidences from previous works (Bowman et al., 2015; Fu et al., 2019b). To verify that our framework is capable of improving text latent diffusion, we have conducted an experiment using two publicly available text datasets: ATIS (Hemphill et al., 1990) and SNIPS (Coucke et al., 2018).

The numbers in this table represent BLEU scores (Papineni et al., 2002), a widely used metric for evaluating text generation models. Higher scores indicate better performance. As the results shown in Table 5, we can see that our framework has significantly improved the text latent diffusion and notably outperformed a strong baseline—Skip Connections—across all datasets and backbones. Therefore, our framework also applies to text data.

**Experiment on image data.** For image data, we provide a detailed discussion in Sec. 4.1 of our paper: Image latent diffusion is rarely affected by posterior collapse due to its non-autoregressive decoder. To confirm this claim in practice, we conduct an experiment comparing latent diffusion with our framework on the widely used CIFAR-10 dataset (Krizhevsky et al., 2009).

The results are shown in Table 6. The numbers in this table represent FID scores (Naeem et al., 2020), a common metric for evaluating image generation models. Lower scores indicate better performance. Our results show that both the baseline model (i.e., KL Annealing) and our framework improve the image latent diffusion to some extent. However, the improvements are minor, suggesting that image models are almost free from posterior collapse.

