# OpenReview forum: "A Study of Posterior Stability for Time-Series Latent Diffusion"
_ICLR.cc/2025/Conference — Submitted to ICLR 2025_

### Official Review · Reviewer_1wS4 · 2024-11-02

**Soundness:** 3
**Presentation:** 3
**Contribution:** 3
**Rating:** 8
**Confidence:** 4

**Summary:**

This work starts from an analysis on the posterior collapse issue of latent diffusion for capturing time series. In particular, the KL-divergence from the standard Gaussian prior to the latent variable distribution approximated using latent diffusion, may reduce the time series latent diffusion to a VAE model. The authors define a dependency measure, which shows that the influences of the latent variables on decoding the observations, will decrease to zero along the diffusion time steps. In particular, as analyzed in the paper, the decoder built upon recurrent neural nets may decode the current observations only using the past observations, and thus leads to the dependency illusion issues. To address the problems, the paper develops a novel framework, in which the authors remove the KL-divergence regularization that causes the posterior collapse, decode the predicting observations using the latent variable sampled at intermediate diffusion steps, and introduce a novel penalty to avoid dependency illusion issues. The final experiments demonstrate the new framework can effectively avoid posterior collapse, and thus achieves superior generation quality, in comparisions to some SOTA time series latent diffusion methods.

**Strengths:**

Novelty: It is the first work to discuss the posterior collapse issue for time series latent diffusion. In particular, the paper introduces the novel dependency measure to quantify how the impacts of the latent variables on the generation of the predicting observations, decrease along the time steps. The authors develop a novel framework, which effectively avoid the dependency illusion issue, and outperforms the related time series latent diffusion models in terms of generation quality.

Clarity: This work clearly illustrates the posterior collapse and dependency illusion issues by plotting the dependency measures over time steps. The most parts of the analysis are clearly presented, and easy-to-follow.

Significance: The work demonstrate a significant issue for latent diffusion being applied to capturing time series data. The introduce dependency measure might be used in quantifying the posterior stability of the other related methods, and thus appears to be crucial.

**Weaknesses:**

The final experiments only demonstrate the compared models on only three dataset, using Wasserstein distance to measure between the true and generated time series. Perhaps the experiments could be enhanced by considering more evaluations metrics?

**Questions:**

I agree on the importance of posterior collapse issue found by the authors. I am wondering how is the generation performance of time series diffusion model in which latent variables have the same dimension as the observations.

---

> ### Author Response · Authors · 2024-11-19
> **Rebuttal**
>
> We thank the reviewer for such kind and professional feedback.
>
> ## Part-1: More datasets and evaluation metrics.
>
> Due to the limited space in the main text, we previously placed the experiment results with two extra time-series datasets (i.e., Retail and Energy) and another evaluation metric (i.e., MMD) in Table 4 of our Appendix F.3. We also explored other data modalities (e.g., texts) in Table 5 and Table 6 of our Appendix F.4.
>
> This part aims to address your concerns in Weakness-1.
> We welcome any further questions you might have.
>
> ## Part-2: Same-dimensional latent variables.
>
> In the case where latent variable $z$ has the same dimension as $X$, the issue of posterior collapse might indeed be mitigated to some extent, though such a high-dimensional latent variable will make the latent diffusion as computationally costly as the vanilla diffusion model.
>
> As you suggested, we conducted an experiment that ran latent diffusion with a latent variable of the same dimension (i.e., 360) as time series. The results are as below.
>
> | **Method**            | **Dimension** | **Backbone**   | **MIMIC** |
> |------------------------|---------------|----------------|-----------|
> | Latent Diffusion       | 64            | Transformer    | 5.02      |
> | Latent Diffusion       | 360           | Transformer    | 3.71      |
> | Our Framework          | 64            | Transformer    | **2.13**  |
> | Latent Diffusion       | 64            | LSTM           | 5.19      |
> | Latent Diffusion       | 360           | LSTM           | 3.82      |
> | Our Framework          | 64            | LSTM           | **2.29**  |
>
> This table shows that **the performance gain achieved by a high-dimensional latent variable is not as significant as that achieved by our framework**. One possible explanation for these results is that the high dimensionality of latent variables makes the diffusion model harder to learn [1, 2], though it can mitigate the issue of posterior collapse.
>
> This part aims to address your concerns in Question-1. We welcome any further questions you might have.
>
> ## Reference
> [1] High-Resolution Image Synthesis with Latent Diffusion Models, CVPR-2022
>
> [2] Score-based Generative Modeling in Latent Space, NeurlPS-2021

---

> > ### Comment · Reviewer_1wS4 · 2024-11-22
> > **RE**
> >
> > Ive read your feedback, and want to thank you. I stick with my score.

---

> > > ### Author Response · Authors · 2024-11-22
> > >
> > > Many thanks for your support, and thank you again for reviewing our paper!

---

### Official Review · Reviewer_aB87 · 2024-11-02

**Soundness:** 2
**Presentation:** 2
**Contribution:** 2
**Rating:** 5
**Confidence:** 3

**Summary:**

This paper aims to address the posterior collapse problem of latent diffusion for time series data. The authors propose a dependency measure method to quantify how posterior collapse happens. And they propose a KL-divergence regularization based method to improve the sensitivity of decoder to the latent variable for time-series latent diffusion.

**Strengths:**

The authors focus on an important issue of latent diffusion, that is posterior collapse, and propose a potential method to quantify the posterior collapse of latent diffusion for time series data.

**Weaknesses:**

1.Regarding to the dependency illusion, you give an example (upper right subfigure of figure 1) to explain. But it’s unclear from figure 1 how you arrive at the conclusion that "Even when the time series is randomly shuffled and thus lacks structural dependencies, the decoder of latent diffusion still heavily relies on input observations (instead of the latent variable) for prediction." Could you clarify how you determine that the decoder "heavily" depends on input observations? Providing a more detailed explanation or additional quantitative evidence would help support this observation.

2.In section 3.1, the definition of posterior collapse seems a general term for all data, not only time series data. In section 3.2, you introduce a dependency measure to demonstrate the occurrence of posterior collapse in time series data. How does this measure specifically address time series data? Would this measure yield the same conclusion if applied to non-time series data?

3.As shown in Figure 4, the dependency of the decoder on the latent variable decreases across both datasets. Although this trend appears improved compared to Figure 2, it would strengthen your findings to compare your method against additional baseline models, rather than only basic latent diffusion.

4.There is a lack of experimental evidence supporting that the proposed dependency measure can accurately assess the impact of the latent variable on the decoder. You should compare your method with other measurement approaches and demonstrate how it outperforms them, providing a more comprehensive validation of its effectiveness.

5.The baselines are not sufficient enough. Only three baselines and all of them are before 2019. Please compare with more the state-of-art works.

6.Reference in this paper seems to be too old. And some of them are repeated. For example, papers in line 573 and line 576 are the same one.

**Questions:**

1.You claim that “when applied to time series data, this framework might suffer from posterior collapse” in the first paragraph. Do you have any evidence to support this claim? Is this phenomenon due to the diminishing influence of the latent variable on the decoder over time steps? How do you justify the decreased dependency correspond to the posterior collapse of latent diffusion for time series data?

2.In section 4.2, you mention that the the variational inference in your framework leads the latent variable to be smooth in its effect on decoder. Is this the reason why your framework can increase the sensitivity of decoder to latent variable?  Can your framework be applied to non-time series data? It seems the proposed method is not specific to time series data.

---

> ### Author Response · Authors · 2024-11-19
> **Rebuttal, Section 0**
>
> Many thanks for your comprehensive and constructive feedback.
>
> ## Part-1: A recap of dependency measures.
>
> We would like to provide a step-by-step review of the key technique introduced in our paper: dependency measures, which might help your understanding and answer related concerns.
>
> ### **1, Motivation of this technique**
>
> A serious consequence of posterior collapse is that the time-series decoder tends to ignore the input latent variable for conditional generation. While this fact is widely recognized in the literature [1,2,3], there remains a lack of a principled method to quantify how much a decoder might neglect the latent variable. The dependency measure was developed in this background, and it will be indispensable for practitioners to diagnose the problem of posterior collapse.
>
> ### **2, Underlying principle (related to your concerns)**
>
> The dependency measure is a typical gradient-based attribution method [4], **with a specific design for time-series generative models, considering the variable length, autoregressive nature, and discrete structure of time series**. The core idea is to measure the sensitivity of a temporal model to its input variable through first-order gradients. The gradient-based attribution itself is principled, with many previous theoretical and empirical works [5,6] that verified its effectiveness.
>
> ### **3, Notations and key properties**
>
> Given a time-series decoder $f$, the global dependency $m_{t,0}$ is a signed measure estimating the dependency of predicting variable $x_t$ on latent variable $z$, while local dependency $m_{t, i}, 0 < i < t$ quantifies such dependency on variable $x_i$.
>
> The measure $m_{t, i}, 0 \le i < t$ is always bounded between $-1$ and $1$, satisfying that $\sum_{0 \le i < t} m_{t, i} = 1$. In typical cases where time series exhibit structural dependencies among variables, $m_{t, i}, 0 \le i < t$ is mostly non-negative, so we can infer:
> -  $m_{t, i} \approx 1$ means that the variable $x_i, i > 0$ or latent variable $z, i = 0$ dominates the attention of decoder $f$;
> - $m_{t, i} \approx 0$ suggests quite the opposite: the decoder $f$ ignores the variable.
>
> ### **4, Use cases (related to your concerns)**
>
> As mentioned in the beginning, the most significant symptom of posterior collapse is that the time-series decoder tends to ignore the input latent variable for conditional generation [1,2,3]. In that situation, based on the properties of dependency measures, the global dependency $m_{t, 0}$ should be close to $0$ for every time step $t$, or soon vanishes with increasing time $t$. In light of this diagnostic logic, **posterior collapse can be asserted when vanishing global dependencies (i.e., $m_{t, 0} \approx 0$ for at least large $t$) are observed**.
>
> On the other hand, based on the property that all measures sum to $1$, one can claim that the decoder heavily depends on input observations under the impact of posterior collapse: high local dependencies $\sum_{1 \le i < t} m_{t, i} = 1 - m_{t, 0} \approx 1$. **This is exactly the case when we explained the upper right subfigure of Fig. 1**.
>
> ### **6, For other data modalities (related to your concerns)**
>
> The dependency measures are specifically designed for time series but can naturally be extended to other types of sequential data (e.g., text). As mentioned in our experiments in Appendix F.4, text Latent Diffusion also seriously suffers from posterior collapse, which only has a minor impact on image Latent Diffusion. Therefore, **dependency measures will yield similar outcomes for text and time-series data, while exhibiting different behavior for image data**.
>
> This part aims to address your related concerns in Weakness-1, 2, 4 and Question-1. We welcome any further questions you might have.

---

> ### Author Response · Authors · 2024-11-19
> **Rebuttal, Section 1**
>
> ## Part-2: More recent baselines and other data modalities.
>
> As you recommended, **we have newly adopted two up-to-date baselines that aimed to address posterior collapse**:
> - One is Inverse Lipschitz Constraint [7] from ICML-2023;
> - The other is Mutual Information Constraint [8] from JMLR-2022.
>
> The experiment results are shown in the below table.
>
> | **Method \ Dataset**                                                          | **MIMIC** | **Earthquakes** |
> |-------------------------------------------------------------------------------|-----------|-----------------|
> | Transformer Latent Diffusion                                                  | 5.02      | 5.91            |
> | Transformer Latent Diffusion w/ Skip Connections                              | 3.75      | 3.69            |
> | Transformer Latent Diffusion w/ Mutual Information Constraint (JMLR-2022)       | 3.59      | 3.85            |
> | Transformer Latent Diffusion w/ Inverse Lipschitz Constraint (ICML-2023)        | 3.01      | 3.42            |
> | Transformer Latent Diffusion w/ Our Framework                                | **2.13**  | **2.49**        |
>
> We can see the two new baselines can indeed mitigate the problem of posterior collapse, though their performance gains are much smaller than our framework. We will include the above new experiment results in the final version.
>
> On the other hand,*we had indeed considered other data modalities (e.g., images). **In Appendix F.4 of our paper, we compared our models with the baselines on text and image datasets**, with experiment results shown in Table 5 and Table 6.
>
> This part aims to address your concerns in Weakness-3, 5 and Question-2.
> We welcome any further questions you might have.
>
> ## Part-3: Other concerns in Weakness-6 and Question-2.
>
> We have introduced 2 new baselines that addressed posterior collapse in our Part-2 answer to you, with other 2 up-to-date time-series generative baselines in our Part-2 answer to Reviewer NiiK. **These 4 new baselines are all from accepted papers in 2023 or 2024**. We will also delete repeatedly cited papers in the final version.
>
> **The key component in our framework that makes the decoder sensitive is the collapse simulation loss** as defined in Eq. (14). The core idea is to apply the diffusion process to simulate a posterior-collapsed latent variable $z$: $P(z|X) \approx P(z)$, and penalize the decoder if it yields high conditional probability $P(X|z)$ in this case. Our experiments in Fig. 4 and Table 1 both verified the effectiveness of this method in practice.
>
> ## References
> [1] Generating Sentences from a Continuous Space, ACL-2016
>
> [2] Cyclical Annealing Schedule: A Simple Approach to Mitigating KL Vanishing, NAACL-2019
>
> [3] Lagging Inference Networks and Posterior Collapse in Variational Autoencoders, ICLR-2019
>
> [4] Axiomatic Attribution for Deep Networks, ICML-2017
>
> [5] A Rigorous Study of Integrated Gradients Method and Extensions to Internal Neuron Attributions, ICML-2022
>
> [6] Guided Integrated Gradients: An Adaptive Path Method for Removing Noise, CVPR-2021
>
> [7] Controlling Posterior Collapse by an Inverse Lipschitz Constraint on the Decoder Network, ICML-2023
>
> [8] Mutual Information Constraints for Monte-Carlo Objectives to Prevent Posterior Collapse Especially in Language Modelling, JMLR-2022

---

> > ### Comment · Reviewer_aB87 · 2024-11-26
> >
> > Thanks for the response and update! These do not affect my original overall review and I keep the original rating.

---

### Official Review · Reviewer_NiiK · 2024-11-04

**Soundness:** 3
**Presentation:** 3
**Contribution:** 2
**Rating:** 5
**Confidence:** 2

**Summary:**

This paper proposes a new approach to establish a stable posterior for time series within the latent diffusion framework. The new approach circumvents the problematic KL-divergence regularization, prevents the posterior collapse, and maintains the influence of the latent variable in the decoder. The authors provide both theoretical and experimental support to their newly proposed framework.

**Strengths:**

The study of the latent diffusion model when applied in the context of time series is a trended topic and super interesting. The authors approach this by defining a proper dependency measure to quantify the problem of posterior collapse of the latent variable, and propose a new framework inspired by re-thinking the design of VAE and autoencoders. The new framework is equipped with new loss functions and regularizations, free from posterior collapse. The discussion comes together with empirical support. Overall, the paper's content is clear and easy to follow.

**Weaknesses:**

1. The experimental results mainly focus on real-world data to demonstrate the sampling benefits of the proposed method. Can the authors conduct synthetic data experiments to interpret and validate the effectiveness of the newly proposed component in the framework (e.g., the introduced loss function or regularization)?

2. The prediction ability of a time series model is critical. Can the authors evaluate the proposed framework in terms of other metrics, such as the predictive MAE, to demonstrate its prediction ability?

3. In addition to point 2, can the author compare with other advanced time-series models? Only compared with the latent diffusion family would not be convincing enough for advertising the model in the time series community.

**Questions:**

N/A

---

> ### Author Response · Authors · 2024-11-19
> **Rebuttal**
>
> We thank the reviewer for such kind and constructive feedback.
>
> ## Part-1: Time-series Generative Models vs Forecasting Models.
>
> Our paper focuses on Latent Diffusion, a type of Generative Models, which are very different from time-series Forecasting Models. **The following table compares the Generative and Forecasting Models, showing their significant difference in evaluation metrics.**
>
> |         | **Time-series Generative Models**                           | **Time-series Forecasting Models**                             |
> |--------------------------|------------------------------------------------------------|----------------------------------------------------------------|
> | **Representative Methods** | Time-series GAN, VAE, Diffusion Models, etc.               | ARIMA, TCNs, Transformer, etc.                                 |
> | **Task Definition**       | Learning a latent representation $z$ of time series $X$, with a map to convert it into time series: $P(X, z)$ | Conditioning on a sequence of observations to predict the next one: $P(x_n \mid x_{n-1}, x_{n-2}, \cdots, x_1)$ |
> | **Evaluation Metrics**    | **Divergence measures (e.g., KL divergence) or Wasserstein distance that measure the distribution gap between the generated $X’$ and real time series $X$** | Some accuracy-like metric $l$ (i.e., mean square error (MSE) or F1) that are defined for the prediction of every observation: $l(x_i, x’_i)$ |
> | **Application Scenarios** | Sensitive Data Anonymization, Data Synthesis for Privacy Protection, Molecular Design, etc. | Stock price prediction, weather forecasting, etc.              |
>
> **Key point from the table:**  Generative Models (e.g., Latent Diffusion) are evaluated by distribution-level metrics (e.g., Wasserstein distance): comparing the generated and real sample distributions, which largely differ from Forecasting Models that are evaluated by observation-level accuracy-like metrics (e.g., MSE). Such a distinction stems from their different model definitions.
>
> **Diverse metrics adopted in our paper:** Previously, We had considered the diversity of evaluation metrics. In Appendix F.3 of our paper, Table 4 compared our models with the baselines in terms of another widely used metric: maximum mean discrepancy (MMD) [1].
>
> This part aims to address your concern in Weakness-2. We welcome any further questions you might have.
>
> ## Part-2: Other advanced baselines, and ablation studies.
>
> As you suggested, **we have additionally adopted two up-to-date time-series baselines for comparison**. One is Frequency Diffusion [2], a (not latent) diffusion-based Generative Model appearing in ICML-2024; The other is Neural STPP [3], a flow-based Generative Model appearing in NeurlPS-2023. The experiment results are shown in the below table.
>
> | **Method \ Dataset**                                  | **MIMIC** | **Earthquakes** |
> |-------------------------------------------------------|-----------|-----------------|
> | Transformer Latent Diffusion (CVPR-2022)             | 5.02      | 5.91            |
> | Neural STPP (NeurIPS-2023)                           | 5.13      | 5.82            |
> | Frequency Diffusion (ICML-2024)                      | 4.56      | 5.07            |
> | Transformer Latent Diffusion w/ Our Framework        | **2.13**  | **2.49**        |
>
> We can see that Latent Diffusion is competitive with up-to-date time-series generative baselines, and it can significantly outperform the baselines with our framework, showing the significance of addressing posterior collapse. **Previously, we had conducted ablation experiments as shown in Table 2 of our Appendix F.1**, verifying the effectiveness of different components (e.g., collapse simulation) in our framework and interpreting the effects of their hyper-parameters.
>
> This part aims to address your concern in Weakness-1, 3.
> We welcome any further questions you might have.
>
> ## References
> [1] Training generative neural networks via maximum mean discrepancy optimization, UAI-2015
>
> [2] Time Series Diffusion in the Frequency Domain, ICML-2024
>
> [3] Automatic Integration for Spatiotemporal Neural Point Processes, NeurlPS-2023

---

> ### Author Response · Authors · 2024-11-29
> **Looking Forward to Your Feedback for Our Rebuttal**
>
> Dear Reviewer NiiK,
>
> We would like to thank you again for your kind and insightful review! As the discussion stage is approaching its extended deadline, we noticed that we have not yet heard from you. ***We look forward to your feedback for our rebuttal***, and we would appreciate any improvement in the rating if your concerns were adequately addressed!
>
> Best Regards,
>
> The Authors

---

### Official Review · Reviewer_h3BG · 2024-11-04

**Soundness:** 2
**Presentation:** 2
**Contribution:** 2
**Rating:** 3
**Confidence:** 4

**Summary:**

This paper addresses the issue of posterior collapse in time-series latent diffusion models, a phenomenon that limits model expressivity by reducing it to a simpler VAE model. The authors introduce a dependency-measure regularization aimed at alleviating this collapse specifically within time-series data. Experiment results on three datasets (WARDS, MIMIC, and Earthquakes) demonstrates initial improvements in preventing posterior collapse over shorter timeframes.

**Strengths:**

--The paper tries to focus on the specific issues of time-dependency collapse in the case of time series data and diffusion models.

--The shuffling experiments help illustrate how a latent variable is not being used strongly throughout all time steps

**Weaknesses:**

--The problem is not sufficiently well motivated.  In particular, the two types of mode collapse which in time series (time-dependent and time-independent) are not discussed.  The reduction to a VAE is only about the elimination of the time-dependent influence.  The impact of this simplification is not sufficiently discussed.

--Moreover, the less expressivity is not shown explicitly to be a bad thing in the context of time series in general.  There are potentially time series which are driven by a static latent process

--Although the dependency measure is well-defined, there is little theoretical analysis exploring its properties and its relationship to the reduction to a VAE model

--There is no analysis of the results showing specifically how the introduced technique solved the problem of mode collapse. Results with good Wasserstein distance do not directly imply that the issue of mode collapse was resolved.

- This paper claims that they are the first to address the posterior collapse problem in latent diffusion for time series, but it really boils down to the old autoencoder problem. And the diffusion model became a redundant module when the input is a standard Gaussian distribution is a simple extension of the problem of autoencoder.

**Questions:**

It’s unclear to me why negative local dependency is bad. The authors claimed that it’s because the previous timestamp’s data may be from a low density region and therefore an outlier. But in case that the actual next value to be decoded should indeed be an extreme value, why is that problematic?

Can you discuss the stability of the training? In cases where we 1) train the diffusion model and the decoder together 2) we require the decoder to decode the time series regardless of which timestamp’s noised version of the latent variable is selected.

---

> ### Author Response · Authors · 2024-11-19
> **Rebuttal, Section 0**
>
> We thank the reviewer for his or her comprehensive and constructive feedback.
>
> ## Part-1: A review of Latent Diffusion, posterior collapse, and our framework
>
> We would like to provide a step-by-step review of the posterior-collapsed latent diffusion and our framework, which might help your understanding and answer related concerns.
>
> ### **1, How does Latent Diffusion samples?**
>
> The sampling process of Latent Diffusion takes two steps:
> - The backward process of a diffusion model incrementally denoises a Gaussian noise $z^L$ into a latent variable $z = z^0$;
> - The decoder $f$ of an autoencoder renders the sampled variable $z$ into time series $X$.
>
> Therefore, as a kind reminder, **it is the last variable $z^0$ of the backward process that controls the time-series decoding**, while all its previous variables $z^i, 1 \le i \le L$, are not involved.
>
> ### **2, Definition of posterior collapse**
>
> Based on the above fact, the posterior collapse in our paper was only defined for the last variable $z^0$ in the backward process, rather than for others $z^i, 1 \le i \le L$. We believe this problem setup is quite appropriate, and **defining "time-dependent and time-independent" sub-problems, as you recommended, might not align with the architecture of Latent Diffusion**, unless the decoder $f$ has access to more than just the last variable of the backward process.
>
> On the other hand, the definition of posterior collapse (i.e., "Problem Formulation” paragraph in Sec. 3.1) in our paper is consistent with many previous works in the literature [1,2,3]. We would appreciate any references with the suggested problem setup, though we believe their applicability to our paper might be limited.
>
> ### **3, The significance of studying posterior collapse in Latent Diffusion**
>
> Latent Diffusion represents one of the state-of-the-art generative models, which is well-known in both academia (e.g., LD4LG [4]) and industry (e.g., Stable Diffusion [5]). Therefore, it is very important to study the potential risks of this advanced architecture, such as posterior collapse.
>
> On the other hand, please note some key findings that are first presented in our paper:
>
> - A fully posterior-collapsed Latent Diffusion is equivalent to a simple VAE, making it a much less capable generative model than even a vanilla diffusion model [6];
> - In cases of partial posterior collapse [3], we introduced a principled method: dependency measure, to quantify the severity of the problem, identifying a previously unknown phenomenon: dependency illusion;
> - As shown in Sec. 4.2 of our paper, the posterior collapse can be perfectly addressed with the diffusion process.
>
> These points clearly indicate that **the problem of posterior-collapsed Latent Diffusion goes far beyond previous research focusing on a simple autoencoder**, calling for a more systematic study. Our work was inspired by this background.
>
> ### **4, How does our framework address the problem?**
>
> The key component in our framework to address posterior collapse is the collapse simulation loss as defined in Eq. (14). The core idea is to apply the diffusion process to simulate a posterior-collapsed latent variable $z$: $P(z|X) \approx P(z)$, and penalize the decoder if it yields high conditional probability $P(X|z)$ with non-informative variable $z$. In other words, the defined loss forced the decoder $f$ to condition on latent variable $z$ for generation, making it informative about time series $X$.
>
> Besides the promising main experiment results measured by Wasserstein distance, **Fig. 1 and Fig. 4 of our paper (i.e., dependency measures over time) showed that the latent variable $z$ in our models maintained a stable control over the decoder$ f$ with increasing time $t$, indicating a non-collapsed posterior $P(z | X)$**.
>
> This part aims to address your concerns in Weakness-1, 2, 4.
> We welcome any further questions you might have.

---

> ### Author Response · Authors · 2024-11-19
> **Rebuttal, Section 1**
>
> ## Part-2: A recap of dependency measures
>
> We would like to provide a brief review of a key technique introduced in our paper: dependency measures, which might help your understanding and answer related concerns.
>
> ### **1, Motivation of this technique**
> A serious consequence of posterior collapse is that the time-series decoder tends to ignore the input latent variable for conditional generation. While this fact is widely recognized in the literature [1,2,3], there remains a lack of a principled method to quantify how much a decoder might neglect the latent variable. The dependency measure was developed in this background, and it will be indispensable for practitioners to diagnose the problem of posterior collapse.
>
> ### **2, How it works, and notations**
>
> The dependency measure is a type of gradient-based attribution [7], **with many previous theoretical and empirical works [8,9] that verified its effectiveness**. The core idea is to measure the sensitivity of a temporal model to its input variable through first-order gradients. Given a time-series decoder $f$, the global dependency $m_{t,0}$ is a signed measure estimating the dependency of predicting variable $x_t$ on latent variable $z$, while local dependency $m_{t, i}, 0 < i < t$ quantifies such dependency on variable $x_i$.
>
> ### **3, Key properties, especially about the negative dependencies**
>
> The measure $m_{t, i}, 0 \le i < t$ is always bounded between $-1$ and $1$, satisfying that $\sum_{0 \le i < t} m_{t, i} = 1$.
>
> About the negative measures, please note:
> - **Both positive and negative measures are valid**, though positivity is more common to see because typical time series exhibit structural dependencies among variables.
> - There are cases where time series are shuffled or get noisy, such that non-positive measures might be observed.
>
> In summary, negative measures do not mean bad, and they are not directly related to posterior collapse except for observing negative global dependency $m_{t, 0}$.
>
> ### **4, Use cases, and relation to less expressivity**
> Regarding the use cases of dependency measure and its relation to less expressivity, please note the following facts from our paper:
>
> - Proposition 3.1 of our paper only considered the cases of a fully collapsed posterior: $P(z | X) = P(z)$, where Latent Diffusion will be as inexpressive as a simple VAE.
> - The dependency measure is a diagnostic tool for the cases of either fully or partially collapsed posterior [3]: $P(z | X) \approx P(z)$, where the global dependency $m_{t, 0}$ will be close to $0$ for every time step $t$, or soon vanishes with increasing time $t$.
>
> Therefore, **The dependency measure extends the applicability of Proposition 3.1 to more scenarios.**
>
> This part aims to address your related concerns in Weakness-3 and Question-1.
> We welcome any further questions you might have.
>
> ## Part-3: Other concerns in Question-2
>
> As mentioned in our Part-1 answer, the decoder of vanilla Latent Diffusion only has access to the last variable of the backward diffusion model: $z^0 = z$. Similarly, in our framework, the latent variable fed into the decoder is sampled from a very small number of variables in the backward process. Appendix F.1 of our paper showed experiment results (i.e., Table 2) about the effect of that number. For jointly training the autoencoder and diffusion model, we found it quite stable in practice.
>
> ## References
> [1] Generating Sentences from a Continuous Space, ACL-2016
>
> [2] Cyclical Annealing Schedule: A Simple Approach to Mitigating KL Vanishing, NAACL-2019
>
> [3] Controlling Posterior Collapse by an Inverse Lipschitz Constraint on the Decoder Network, ICML-2023
>
> [4] Latent Diffusion for Language Generation, NeurlPS-2023
>
> [5] High-Resolution Image Synthesis with Latent Diffusion Models, CVPR-2022
>
> [6] A Variational Perspective on Diffusion-Based Generative Models and Score Matching, NeurlPS-2021
>
> [7] Axiomatic Attribution for Deep Networks, ICML-2017
>
> [8] A Rigorous Study of Integrated Gradients Method and Extensions to Internal Neuron Attributions, ICML-2022
>
> [9] Guided Integrated Gradients: An Adaptive Path Method for Removing Noise, CVPR-2021

---

> > ### Comment · Reviewer_h3BG · 2024-11-23
> >
> > I appreciate the authors' detailed feedback to my reviews. While it helps to reiterate the claimed contributions, I am not fully convinced with the argument and therefore would maintain my original score.

---

> ### Author Response · Authors · 2024-11-23
>
> We thank the reply from the reviewer. Below is a point-by-point summary highlighting how our rebuttal tried to address your concerns:
> - "1, How does Latent Diffusion samples?" of Part-1 clarified that time-series decoding is only controlled by the last backward latent variable $z = z^0$ ***(instead of any process), so your concern about "There are potentially time series which are driven by a static latent process" does not apply to the architecture of Latent Diffusion***. *In other words, the virtual time steps in diffusion models have nothing to do with the time steps in time-series data*;
> - "2, Definition of posterior collapse" of Part- clarified that ***our definition of posterior collapse is consistent with many previous works, and we believe that your concern about "time-dependent and time-independent mode collapses" does not fit the problem setting of our paper***.  We would also appreciate any references you could provide;
> - ”3, The significance of studying …" of Part-1 aimed to address your concern about "how the introduced technique solved the problem of mode collapse”;
> - "4, How does our framework address…" of Part-1 aimed to address your concern about “how the introduced technique solved the problem of mode collapse”;
> - ”2, How it works ..." and "3, Key properties … negative dependencies" of Part-2 aimed to address your concerns about "there is little theoretical analysis exploring its properties" and "negative local dependency”. We reviewed the key properties of dependency measures and clarified about negative dependencies;
> - "4, … relation to less expressivity" of Part-2 aimed to address your concern about “its relationship to the reduction to a VAE model”.
>
> A kind reminder is that *your review repeatedly referred to "mode collapse", instead of the posterior collapse studied by our paper*. ***These two terminologies have very distinct definitions in the context of generative models***. We are concerned that there might be some misunderstandings that we could further address.
>
> Thank you again for your reply, and we are looking forward to the opportunity to further clarify any of your concerns that have been not addressed well.

---

### Official Review · Reviewer_QHVr · 2024-11-04

**Soundness:** 2
**Presentation:** 3
**Contribution:** 2
**Rating:** 6
**Confidence:** 3

**Summary:**

The paper investigates the issue of posterior collapse in latent diffusion models for time series data, where the latent variable becomes ineffective in influencing the model’s output. The authors propose a dependency measure to quantify how much the decoder relies on the latent variable, highlighting not only posterior collapse but also a related phenomenon termed dependency illusion. Then the paper introduces a new framework to address these issues by removing KL-divergence regularization and enhancing the decoder’s sensitivity to the latent variable, improving posterior stability. Experiments demonstrate that the proposed method achieves better performance than standard latent diffusion models with posterior collapse mitigation techniques across various time series datasets.

**Strengths:**

- The paper addresses a previously underexplored issue in time series diffusion models—posterior collapse—which has primarily been studied in variational autoencoders (VAEs) but not in the context of diffusion models for time series.
- The dependency measure provides an insightful tool for quantifying the decoder’s reliance on the latent variable. This measure enables detection of both posterior collapse and dependency illusion, offering valuable diagnostic capabilities for latent-variable models.
- The approach aligns with the paper’s theoretical objectives, yielding meaningful performance improvements.

**Weaknesses:**

- The empirical evaluation lacks comparisons with stable time series models that naturally avoid posterior collapse, such as ARIMA, RNNs, LSTMs, transformers, and temporal convolutional networks. Including these baselines would provide context on whether the proposed framework offers advantages beyond mitigating posterior collapse. The author also did not compare with recent baselines for time series, which are diffusion-based. Please check papers published in NeurIPS/ICLR/ICML in the past two years.

- The paper references Bowman et al. (2016) to support claims about posterior collapse in latent-variable models for time series, which may be outdated. This raises questions about whether latent diffusion models represent the current state of the art in time series modeling. Comparing the approach with recent state-of-the-art time series methods would strengthen the justification for the proposed framework.

- Although the datasets used are realistic, the paper does not discuss broader real-world applications or scenarios where posterior stability is crucial, such as in anomaly detection or real-time forecasting. Adding context on practical use cases would clarify the framework’s relevance.

**Questions:**

- Could the authors include comparisons with recent state-of-the-art time series models, such as ARIMA, LSTMs, transformers, and TCNs, which are naturally robust against posterior collapse? This would contextualize the proposed method’s advantages relative to stable baselines.

- Could the authors provide clearer definitions or examples for terms like dependency illusion and posterior collapse in the context of latent diffusion models? A simplified explanation would improve accessibility.

- Are there specific real-world applications, such as anomaly detection or real-time forecasting, where this framework would be particularly useful? A discussion of practical use cases would strengthen the framework’s relevance.

---

> ### Author Response · Authors · 2024-11-19
> **Rebuttal, Section 0**
>
> We thank the reviewer for providing such comprehensive and constructive feedback.
>
> ## Part-1: Time-series Generative Models vs Forecasting Models.
>
> Our paper focuses on **Latent Diffusion, a type of Generative Models different from your recommended models (e.g.,  ARIMA and TCNs), which belong to time-series Forecasting Models** and are indeed free from posterior collapse. The following table compares the two classes of models, showing the use cases of our paper.
>
> | | Time-series Generative Models                                   | Time-series Forecasting Models                                |
> |-----|------------|-----------|
> | **Representative Methods** | Time-series GAN [1], VAE, Diffusion Models, etc.             | ARIMA, TCNs, Transformer, etc.                                |
> | **Task Definition** | Learning a latent representation $z$ of time series $X$, with a map to convert it into time series: $P(X, z) $  | Conditioning on a sequence of observations to predict the next one: $P(x_{n} \mid x_{n-1}, x_{n-2}, \cdots, x_1) $ |
> | **Main Concerns** | **Posterior collapse**, fairness [2], memorization [3], etc. | Model expressiveness, autoregressive modeling, graph neural networks, etc. |
> | **Application Scenarios**    | Sensitive Data Anonymization [10], Data Synthesis for Privacy Protection [11], Molecular Design [12], etc. | Stock price prediction, weather forecasting, etc. |
>
>
> **Key points from the table:** Time-series Forecasting Models (e.g., ARIMA) are without a key component of Generative Models (e.g., Latent Diffusion):  latent variable z, which might incur posterior collapse. For this reason, well-performing Forecasting Models do not suffer from posterior collapse, though we can see that Generative Models also have their unique values in real-world applications (e.g., Data Synthesis and Drug Discovery).
>
> **Advanced time-series architectures in our models:** On the other hand, we built the time-series decoder of our Generative Models with either modern Transformer or LSTM (see Table 1 of our paper), which are both your recommendations. As indicated above, while those architectures are without posterior collapse, the latent variable z that initializes them is the root cause incurring the problem.
>
> This part aims to address your concerns in Weakness-1, 3 and Question-1, 3.
> We welcome any further questions you might have.
>
> ## Part-2: Latent Diffusion is the state-of-the-art, with more recent baselines adopted.
>
> Our paper is based on Latent Diffusion that first appeared in CVPR-2022, representing one of the most advanced architectures of Generative Models. **There are many very recent papers that focused on time-series Latent Diffusion**, or even more broadly: sequence data with Latent Diffusion. For example, TimeLDM [7], LD4LG [8], and AudioLDM [9] that appear in NeurlPS-2023, ICML-2023, and the arXiv months ago. Therefore, from the recent literature, we believe that Latent Diffusion stands as a state-of-the-art time-series Generative Model.
>
> As you recommended, **we have additionally adopted two up-to-date time-series baselines for comparison**:
> 1. One is Frequency Diffusion [10], a (not latent) diffusion-based Generative Model appearing in ICML-2024;
> 2. The other is Neural STPP [11], a flow-based Generative Model appearing in NeurlPS-2023.
>
> The experiment results are shown in the below table.
>
>  | Method / Dataset                                              | MIMIC  | Earthquakes |
> |---------------------------------------------------------------|--------|-------------|
> | Transformer Latent Diffusion (CVPR-2022)                      | 5.02   | 5.91        |
> | Neural STPP (NeurIPS-2023)                                    | 5.13   | 5.82        |
> | Frequency Diffusion (ICML-2024)                               | 4.56   | 5.07        |
> | Transformer Latent Diffusion w/ Our Framework                 | **2.13** | **2.49**    |
>
> We can see that **vanilla Latent Diffusion achieved competitive performances to the up-to-date baselines** (e.g. Neural STPP), showing that it is still very advanced. In particular, **our framework significantly improves Latent Diffusion, with performances notably outperforming the baselines**, indicating that posterior collapse is indeed a performance bottleneck of Time-series Latent Diffusion. The above empirical results further confirmed that Latent Diffusion is a state-of-the-art time-series Generative Model, and verified the importance of addressing posterior collapse.
>
> This part aims to address your concerns in Weakness-1, 2 and Question-1. We welcome any further questions you might have.

---

> ### Author Response · Authors · 2024-11-19
> **Rebuttal, Section 1**
>
> ## Part-3: Explanations of some terminologies.
>
> We prepared the below table that re-explains and connects some key terminologies appearing in our paper.
>
> | | **Posterior Collapse**                                         | **Dependency Measure**                                      | **Dependency Illusion**                                        |
> |-----|--------|--------|--------|
> | **Definition in the paper**                                   | Paragraph "problem formulation" in Sec. 3.1                | Definition 3.2 in Sec. 3.2                                      | Paragraph "Insightful results" in Sec. 3.3                       |
> | **Explanation**   | The posterior $P(z \mid X)$ reduces to the prior $P(z),$ indicating that the latent variable $z$ is not informative about data $X$ | A principled method to quantify the impact of observation $x_i$ or latent variable $z$ to predict observation $x_j, j > i$ | Different observations $x_i, x_j, 1 \le i < j \le n$ in time series $X$ are totally or almost independent, but the decoder $f$ still highly relies on $x_i$ to predict $x_j$ (e.g., high dependency measure $m_{j,i}$) |
> | **Newly Introduced?**                                          | No                                                          | Yes                                                           | Yes                                                            |
> | **Negative impacts on Latent Diffusion**                       | Making Latent Diffusion less expressive as a Generative Model (Proposition 3.1) and reducing the sensitivity of decoder $f$ to latent variable $z$ (Sec. 3.3) | N/A                                                           |  Another implication of posterior collapse: the decoder $f$ incorrectly captures the relationships between different observations, which is not desired for conditional generation $P(X \mid z)$ |
> | **Related experiments in the main text**                       | Fig. 1, Fig. 2, Fig. 4, Table 1                              | Fig. 1, Fig. 2, Fig. 4                                         | Fig. 1, Fig. 2, Fig. 4, Table 1                                  |
>
> This table aims to answer your Question-2, and we will include it in the final version for better clarity. We welcome any further questions you might have.
>
> ## References
> [1] Time-series Generative Adversarial Networks, NeurlPS-2019
>
> [2] On Measuring Fairness in Generative Models, NeurlPS-2023
>
> [3] On Memorization in Probabilistic Deep Generative Models, NeurlPS-2021
>
> [4] Anonymization Through Data Synthesis Using Generative Adversarial Networks, IEEE-2020
>
> [5] Data Synthesis based on Generative Adversarial Networks, VLDB-2018
>
> [6] Equivariant Diffusion for Molecule Generation in 3D, ICML-2022
>
> [7] TimeLDM: Latent Diffusion Model for Unconditional Time Series Generation, arXiv-2024
>
> [8] Latent Diffusion for Language Generation, NeurlPS-2023
>
> [9] AudioLDM: Text-to-Audio Generation with Latent Diffusion Models, ICML-2023
>
> [10] Time Series Diffusion in the Frequency Domain, ICML-2024
>
> [11] Automatic Integration for Spatiotemporal Neural Point Processes, NeurlPS-2023

---

> > ### Comment · Reviewer_QHVr · 2024-11-25
> > **Thanks for the reply.**
> >
> > Dear Authors, thanks for the reply. My concerns are addressed. Hence, I now increase my score.

---

> > > ### Author Response · Authors · 2024-11-25
> > >
> > > Thank you for your positive feedback, and thanks again for your great efforts in reviewing our paper!

---

### Official Review · Reviewer_wREh · 2024-11-05

**Soundness:** 2
**Presentation:** 2
**Contribution:** 3
**Rating:** 5
**Confidence:** 5

**Summary:**

This paper addresses the problem of posterior collapse in latent diffusion models, specifically when applied to time series data. The authors provide a systematic analysis of this issue, showing that posterior collapse can reduce the expressiveness of latent diffusion to that of a variational autoencoder (VAE). They introduce a novel dependency measure to quantify the impact of latent variables on the generation process and identify a phenomenon called dependency illusion when time series data are shuffled. Building on these insights, the authors propose a new framework that eliminates the KL-divergence regularization, permits an expressive prior distribution, and ensures the decoder remains sensitive to the latent variable. Extensive experiments demonstrate that this framework avoids posterior collapse and significantly improves time series generation.

**Strengths:**

1.	The introduction of dependency measures to diagnose and address posterior collapse is both novel and insightful, providing a fresh perspective on an important issue within latent diffusion models.
2.	The paper offers a solid theoretical foundation for the analysis of posterior collapse, and the proposed framework is well-motivated by both theoretical insights and empirical observations.
3.	The proposed framework demonstrates significant improvements in the performance of time-series generation models, effectively addressing a key limitation in existing approaches.

**Weaknesses:**

1.	While the paper presents strong results for time-series data, it lacks a detailed discussion on the generalizability of the approach to other data modalities, such as images or text. Including a brief exploration or discussion of potential extensions could further enhance the contribution.
2.	The experimental details, including specific configurations for baselines and the selection of hyperparameters, are not fully elaborated in the main text. Providing more comprehensive explanations in these areas would improve the paper’s clarity and reproducibility.
3.	Although the results are promising, some of the visualizations could be made more intuitive, particularly for readers unfamiliar with latent diffusion models. Additionally, converting the figures to vector graphics would significantly improve their quality, as several of the current images appear blurry and lack sharpness, which makes interpretation more difficult. Enhancing the clarity of the figures would improve the overall presentation of the paper.

**Questions:**

1.	Could the authors clarify how the dependency measure scales with longer time-series datasets? Does the framework handle large datasets efficiently?
2.	Have the authors considered extending this approach to other data types beyond time series? If so, how might the framework need to be adapted?
3.	Is there a specific reason for not including additional baselines, such as non-latent diffusion models, for comparison in the empirical section?

---

> ### Author Response · Authors · 2024-11-19
> **Rebuttal**
>
> We thank the reviewer for his or her kind and comprehensive feedback.
>
> ## Part-1: More baselines, and other data modalities.
>
> As you recommended, **we have additionally adopted two up-to-date baselines for comparison**:
> 1. One is Frequency Diffusion [1], a (not latent) diffusion-based Generative Model appearing in ICML-2024;
> 2. The other is Neural STPP, a flow-based Generative Model appearing in NeurlPS-2023.
>
> The experiment results are shown in the below table.
>
> | Method / Dataset                                              | MIMIC  | Earthquakes |
> |---------------------------------------------------------------|--------|-------------|
> | Transformer Latent Diffusion (CVPR-2022)                      | 5.02   | 5.91        |
> | Neural STPP (NeurIPS-2023)                                    | 5.13   | 5.82        |
> | Frequency Diffusion (ICML-2024)                               | 4.56   | 5.07        |
> | Transformer Latent Diffusion w/ Our Framework                 | **2.13** | **2.49**    |
>
> We can see that Latent Diffusion is competitive with up-to-date time-series generative baselines, and it can significantly outperform the baselines with our framework, showing the significance of addressing posterior collapse.
>
> On the other hand, **we had indeed considered other data modalities (e.g., images)**. In Appendix F.4 of our paper, we compared our models with the baselines on text and image datasets, with experiment results shown in Table 5 and Table 6.
>
> This part aims to address your concerns in Weakness-1 and Question-2, 3. We welcome any further questions you might have.
>
> ## Part-2: Scalability of dependency measures.
>
> From Eq. (10) of our paper, we can see that the computational complexity of a dependency measure is $O(NM)$, where $N$ is the number of Monte Carlo samples and $M$ is the length of time series. Therefore, the computational cost of the dependency measure linearly grows with the increasing sentence length, and this linear complexity can be further optimized with parallel GPU computation. In practice, a set of 10000 time series with an average length of 30 cost us only about 5min on a single GPU device. In summary, the dependency measure scales to large and long time-series datasets well.
>
> This part aims to address your concern in Question-1. We welcome any further questions you might have.
>
> ## Part-3: Other concerns about experiments.
>
> As mentioned at the beginning of our Experiments section (i.e., Sec. 6), we move the Experiment Details section to Appendix E of our paper, due to the limited space in the main text. We will relocate that section and improve the graph quality as you suggested in the final version.
>
> ## References
>
> [1] Time Series Diffusion in the Frequency Domain, ICML-2024
>
> [2] Automatic Integration for Spatiotemporal Neural Point Processes, NeurlPS-2023

---

> > ### Comment · Reviewer_wREh · 2024-11-25
> > **Response to authors**
> >
> > Thanks the authors' patient feedback.

---

> > > ### Author Response · Authors · 2024-11-25
> > >
> > > Thank you very much for your reply! I understand that you are currently occupied with a large volume of review tasks. If possible, we would appreciate it if you could let us know whether our previous rebuttal has addressed your concerns.

---

> > > > ### Comment · Reviewer_wREh · 2024-12-03
> > > > **One more question**
> > > >
> > > > I completely agree that posterior collapse is a critical issue in diffusion models. However, a key concern is whether this problem is uniquely prominent or particularly evident in the time series domain.

---

> > > > > ### Author Response · Authors · 2024-12-03
> > > > >
> > > > > Glad to receive your new question! Our answer is yes: please refer to Appendix F.3 of our paper, showing that both text and time-series generative models were prone to posterior collapse, which was not the case for images. Therefore, the problem is prominent in typical sequential data, including time series. ***We hope our previous rebuttal had sufficiently addressed your concerns, and if so, we would greatly appreciate it if you could consider improving your rating.***

---

### Author Response · Authors · 2024-11-25
**Looking Forward to Further Feedback**

Dear Reviewers,

Thank you for your comprehensive and constructive reviews, including some latest feedback.

With the discussion stage nearing its conclusion, we look forward to hearing from all other reviewers.

Thank you, and have a wonderful day!

Best Regards,

The Authors

---

### Meta-Review · Area_Chair_jZ7u · 2024-12-19

**Metareview:**

This paper proposed a metric to measure the posterior collapse and introduced a notion of the posterior collapse based on it. They then define an enhanced framework which can alleviate the posterior collapse problem. I found that their posterior collapse is different from the model collapse and Reviewer h3BG is incorrect to some degree.

In conjunction with the reviewers' concerns regarding the comparison with more baselines, however, I, myself, also have some concerns.

1. They need to more carefully analyze what causes the posterior collapse. For instance, LSTMs and Transformers have oversmoothing and oversquashing problems and thereby, their latent vectors are limited in capturing all information in a long sequence. Isn't the posterior collapse by the low-capacity encoder, e.g., LSTMs and Transformers?

2. Your new framework incurs more interactions and I agree that it can somehow address the problem. It would be nice if you can show that enhancing the encoder/decoder is not sufficient and your framework is needed. There are several papers on resolving the oversmoothing and oversquashing problems of RNNs and Transformers.

3. After solving the above two questions, I recommend that you put more baselines since there are many time series synthesis methods after TimeGAN. Some of them are diffusion-based methods for time series synthesis.

All in all, I strongly encourage the authors improve the paper one more time. I think they can have a decent chance next time since their problem definition is new.

**Additional Comments On Reviewer Discussion:**

The authors provided good messages and however, most reviewers feel that this paper is slightly below the acceptance threshold.

---

### Decision · Program_Chairs · 2025-01-22

Reject